# A BEST-OF-BOTH-WORLDS ALGORITHM FOR MDPS WITH LONG-TERM CONSTRAINTS

## ABSTRACT

We study *online learning* in episodic *constrained Markov decision processes* (CMDPs), where the goal of the learner is to collect as much reward as possible over the episodes, while guaranteeing that some *long-term* constraints are satisfied during the learning process. Rewards and constraints can be selected either *stochastically* or *adversarially*, and the transition function is *not* known to the learner. While online learning in classical (unconstrained) MDPs has received considerable attention over the last years, the setting of CMDPs is still largely unexplored. This is surprising, since in real-world applications, such as, *e.g.*, autonomous driving, automated bidding, and recommender systems, there are usually additional constraints and specifications that an agent has to obey during the learning process. In this paper, we provide the first *best-of-both-worlds* algorithm for CMDPs with long-term constraints. Our algorithm is capable of handling settings in which rewards and constraints are selected either stochastically or adversarially, without requiring any knowledge of the underling process. Moreover, our algorithm matches state-of-the-art regret and constraint violation bounds for settings in which constraints are selected stochastically, while it is the first to provide guarantees in the case in which they are chosen adversarially.

## 1 INTRODUCTION

The framework of *Markov decision processes* (MDPs) (Puterman, 2014) has been extensively employed to model sequential decision-making problems. In *reinforcement learning* (RL) (Sutton & Barto, 2018), the goal is to learn an optimal policy for an agent interacting with an environment modeled as an MDP. A different line of work (Even-Dar et al., 2009; Neu et al., 2010) is concerned with problems in which an agent interacts with an unknown MDP with the goal of guaranteeing that the overall reward achieved during the learning process is as much as possible. This approach is more akin to *online learning* problems, and it is far less investigated than classical RL approaches.

In real-world applications, there are usually additional constraints and specifications that an agent has to obey during the learning process, and these cannot be captured by the classical definition of MDP. For instance, autonomous vehicles must avoid crashing while navigating (Wen et al., 2020; Isele et al., 2018), bidding agents in ad auctions are constrained to a given budget (Wu et al., 2018; He et al., 2021), while recommender systems should *not* present offending items to users (Singh et al., 2020). In order to model such features of real-world problems, Altman (1999) introduced *constrained* MDPs (CMDPs) by extending classical MDPs with cost constraints which the agent has to satisfy.

We study *online learning* in episodic CMDPs in which the agent is subject to *long-term* constraints. In such a setting, the goal of the agent is twofold. On the one hand, the agent wants to minimize their (cumulative) *regret*, which is how much reward they lose compared to that achieved by playing the same best-in-hindsight, constraint-satisfying policy in all the episodes. On the other hand, while the agent is allowed to violate the constraints in a given episode, they want that the (cumulative) *constraint violation* over all the episodes stays under control by growing sublinearly in the number of episodes. Long-term constraints can naturally model many features of real world problems, such as, *e.g.*, budget depletion in automated bidding (Balseiro & Gur, 2019; Gummadi et al., 2012).

All the existing works studying online learning problems in CMDPs with long-term constraints address settings in which the constraints are selected stochastically according to an unknown (stationary) probability distribution. While these works address both the case where the rewards are

stochastic (see, *e.g.*, (Zheng & Ratliff, 2020; Efroni et al., 2020)) and the one in which they are chosen adversarially (see, *e.g.*, (Wei et al., 2018; Qiu et al., 2020)), to the best of our knowledge there is no work addressing settings with adversarially-selected constraints. Some works (see, *e.g.*, (Ding & Lavaei, 2023; Wei et al., 2023)) consider the case in which rewards and constraints are non-stationary, assuming that their variation is bounded. However, such results are *not* applicable to general settings with adversarial constraints. A detailed discussion of related works can be found in Appendix A.

In this paper, we pioneer the study of CMDPs in which the constraints are selected adversarially. In doing so, we introduce a *best-of-both-worlds* algorithm that provides optimal (in the number of episodes $T$) regret and constraint violation bounds when rewards and constraints are selected either *stochastically* or *adversarially*, without requiring any knowledge of the underling process. While algorithms of this kind have been recently introduced in online learning settings (see, *e.g.*, (Liakopoulos et al., 2019)), to the best of our knowledge ours is the first of its kind in CMDPs.

When the constraints are selected stochastically, we show that our algorithm provides $\tilde{\mathcal{O}}(\sqrt{T})$ cumulative regret and constraint violation when a suitably-defined Slater-like condition concerning the satisfiability of constraints is satisfied. Moreover, whenever such a condition does *not* hold, our algorithm still ensures $\tilde{\mathcal{O}}(T^{3/4})$ regret and constraint violation. Instead, whenever the constraints are chosen adversarially, our analysis revolves around a parameter $\rho$ which is related to our Slater-like condition, and in particular to the "margin" by which it is possible to strictly satisfy the constraints. Indeed, under adversarial constraints, Mannor et al. (2009) show that it is impossible to simultaneously achieve sublinear regret and sublinear cumulative constraint violation. We prove that our algorithm achieves no-$\alpha$-regret with $\alpha = \rho/(1+\rho)$, while guaranteeing that the cumulative constraint violation is sublinear in the number of episodes. This matches the regret guarantees derived for other best-of-both-worlds algorithms in (non-sequential) online learning settings (Castiglioni et al., 2022a;b). Moreover, differently from previous works on online learning with adversarial constraints, we also relax the strong assumption that the algorithm has to know the value of the feasibility parameter $\rho$. A summary of our contributions compared to those of prior works is reported in Table 1.

| | adv. rewards | adv. constraints | unknown $\rho$ | w/o. Slater | MDPs |
|---|:---:|:---:|:---:|:---:|:---:|
| (Efroni et al., 2020) | ✗ | ✗ | ✓ | ✓ | ✓ |
| (Qiu et al., 2020) | ✓ | ✗ | ✓ | ✗ | ✓ |
| (Castiglioni et al., 2022b) | ✓ | ✓ | ✗ | ✓ | ✗ |
| (Wei et al., 2023) | ✗[†] | ✗[†] | ✓ | ✗ | ✓ |
| Our Work | ✓ | ✓ | ✓ | ✓ | ✓ |

Table 1: Comparison of our work and the state-of-the-art. We group together previous works that provide similar guarantees. For each group, we only cite the most recent paper. The third column concerns the possibility of learning without the knowledge of the parameter $\rho$, while the fourth one specifies if the algorithm is capable of learning when the parameter $\rho$ is arbitrarily small. [†] These works do *not* apply to general adversarial settings, but only to settings with bounded non-stationarity.

## 2 PRELIMINARIES

### 2.1 CONSTRAINED MARKOV DECISION PROCESSES

We study *episodic constrained* MDPs (Altman, 1999), which we call CMDPs for short and are defined as tuples $M = \left( X, A, P, \{r_t\}_{t=1}^T, \{G_t\}_{t=1}^T \right)$, where:

- $T$ is a number of episodes, with $t \in [T]$ denoting a specific episode.[1]

- $X$ and $A$ are the finite state and action spaces, respectively.

- $P : X \times A \times X \to [0,1]$ is the transition function, where, for ease of notation, we denote by $P(x'|x,a)$ the probability of going from state $x \in X$ to $x' \in X$ by taking action $a \in A$.

---

[1]We denote with $[a .. b]$ the set of all consecutive integers from $a$ to $b$, while $[b] = [1 .. b]$.

- $\{r_t\}_{t=1}^T$ is a sequence of vectors describing the rewards at each episode $t \in [T]$, namely $r_t \in [0, 1]^{|X \times A|}$. We refer to the reward of a specific state-action pair $x \in X, a \in A$ for an episode $t \in [T]$ as $r_t(x, a)$. Rewards may be *stochastic*, in that case $r_t$ is a random variable distributed according to a distribution $\mathcal{R}$ for every $t \in [T]$, or chosen by an *adversary*.

- $\{G_t\}_{t=1}^T$ is a sequence of constraint matrices describing the $m$ *constraint* violations at each episode $t \in [T]$, namely $G_t \in [-1, 1]^{|X \times A| \times m}$, where non-positive violation values stand for satisfaction of the constraints. For $i \in [m]$, we refer to the violation of the $i$-th constraint for a specific state-action pair $x \in X, a \in A$ at episode $t \in [T]$ as $g_{t,i}(x, a)$. Constraint violations may be *stochastic*, in that case $G_t$ is a random variable distributed according to a probability distribution $\mathcal{G}$ for every $t \in [T]$, or chosen by an *adversary*.

---

**Algorithm 1** Learner-environment Interaction

---

1: **for** $t = 1, \dots, T$ **do**
2:     $r_t$ and $G_t$ are chosen *stochastically* or *adversarially*
3:     the learner chooses a policy $\pi_t : X \times A \to [0, 1]$
4:     the state is initialized to $x_0$
5:     **for** $k = 0, \dots, L - 1$ **do**
6:         the learner plays $a_k \sim \pi_t(\cdot|x_k)$
7:         the environment evolves to $x_{k+1} \sim P(\cdot|x_k, a_k)$
8:         the learner observes $x_{k+1}$
9:     **end for**
10:     the learner is revealed $r_t, G_t$
11: **end for**

---

W.l.o.g., in this work we consider *loop-free* CMDPs. Formally, this means that $X$ is partitioned into $L$ layers $X_0, \dots, X_L$ such that the first and the last layers are singletons, *i.e.*, $X_0 = \{x_0\}$ and $X_L = \{x_L\}$, and that $P(x'|x, a) > 0$ only if $x' \in X_{k+1}$ and $x \in X_k$ for some $k \in [0 .. L - 1]$. Notice that any episodic CMDP with horizon $L$ that is *not* loop-free can be cast into a loop-free one by suitably duplicating the state space $L$ times, *i.e.*, a state $x$ is mapped to a set of new states $(x, k)$, where $k \in [0 .. L]$.

The learner chooses a *policy* $\pi : X \times A \to [0, 1]$ at each episode, defining a probability distribution over actions at each state. For ease of notation, we denote by $\pi(\cdot|x)$ the probability distribution for a state $x \in X$, with $\pi(a|x)$ denoting the probability of action $a \in A$. Algorithm 1 depicts the interaction between the learner and the environment in a CMDP. Furthermore, we assume that the learner knows $X$ and $A$, but they do *not* know anything about $P$.

## 2.2 Occupancy measures

Next, we introduce the notion of *occupancy measure* (Rosenberg & Mansour, 2019a). Given a transition function $P$ and a policy $\pi$, the occupancy measure $q^{P,\pi} \in [0, 1]^{|X \times A \times X|}$ induced by $P$ and $\pi$ is such that, for every $x \in X_k$, $a \in A$, and $x' \in X_{k+1}$ with $k \in [0 .. L - 1]$:

$$q^{P,\pi}(x, a, x') = \Pr[x_k = x, a_k = a, x_{k+1} = x'|P, \pi]. \tag{1}$$

Moreover, we also define:

$$q^{P,\pi}(x, a) = \sum_{x' \in X_{k+1}} q^{P,\pi}(x, a, x'), \tag{2}$$

$$q^{P,\pi}(x) = \sum_{a \in A} q^{P,\pi}(x, a).$$

Then, we can introduce the following lemma, which characterizes *valid* occupancy measures.

**Lemma 1** (Rosenberg & Mansour (2019b)). *For every $q \in [0, 1]^{|X \times A \times X|}$, it holds that $q$ is a valid occupancy measure of an episodic loop-free MDP if and only if, for every $k \in [0 .. L - 1]$, the following three conditions hold:*

$$\begin{cases} \sum_{x \in X_k} \sum_{a \in A} \sum_{x' \in X_{k+1}} q(x, a, x') = 1 \\ \sum_{a \in A} \sum_{x' \in X_{k+1}} q(x, a, x') = \sum_{x' \in X_{k-1}} \sum_{a \in A} q(x', a, x) \quad \forall x \in X_k \\ P^q = P \end{cases}$$

*where $P$ is the transition function of the MDP and $P^q$ is the one induced by $q$ (see Equation (3)).*

Notice that any valid occupancy measure $q$ induces a transition function $P^q$ and a policy $\pi^q$ as:

$$P^q(x'|x,a) = \frac{q(x,a,x')}{q(x,a)}, \quad \pi^q(a|x) = \frac{q(x,a)}{q(x)}. \tag{3}$$

### 2.3 OFFLINE CMDPs OPTIMIZATION

We define the parametric linear program $\text{LP}_{r,G}$ (4) with parameters $r$ and $G$ as follows:

$$\text{OPT}_{r,G} := \begin{cases} \max_{q \in \Delta(M)} & r^\top q \\ \text{s.t.} & G^\top q \le \underline{0}, \end{cases} \tag{4}$$

where $q \in [0,1]^{|X \times A|}$ is the occupancy measure vector whose values are defined in Equation (2), $\Delta(M)$ is the set of valid occupancy measures, $r$ is the reward vector, and $G$ is the constraint matrix. Furthermore, we introduce the following condition.

**Condition 1.** *Given a constraint matrix $G$, the Slater's condition holds when there is a strictly feasible solution $q^\diamond$ such that $G^\top q^\diamond < \underline{0}$.*

Then, we define the Lagrangian function for Problem (4).

**Definition 1** (Lagrangian function)**.** *Given a reward vector $r$ and a constraint matrix $G$, the Lagrangian function $\mathcal{L}_{r,G} : \Delta(M) \times \mathbb{R}_{\ge 0}^m \to \mathbb{R}$ of Problem (4) is defined as:*

$$\mathcal{L}_{r,G}(q,\lambda) := r^\top q - \lambda^\top (G^\top q).$$

It is well known (see, *e.g.*, (Altman, 1999)) that strong duality holds for CMDPs assuming Slater's condition. Therefore, we have that the following corollary holds.

**Corollary 1.** *Given a reward vector $r$ and a constraint matrix $G$ such that the Slater's condition holds, we have:*

$$\text{OPT}_{r,G} = \min_{\lambda \in \mathbb{R}_{\ge 0}^m} \max_{q \in \Delta(M)} \mathcal{L}_{r,G}(q,\lambda) = \max_{q \in \Delta(M)} \min_{\lambda \in \mathbb{R}_{\ge 0}^m} \mathcal{L}_{r,G}(q,\lambda).$$

Notice that the min-max problem in Corollary 1 corresponds to the optimization problem associated with a zero-sum Lagrangian game.

### 2.4 CUMULATIVE REGRET AND CONSTRAINT VIOLATION

We introduce the notion of cumulative regret and cumulative constraint violation up to episode $T$.

The cumulative regret is defined as $R_T := T\,\text{OPT}_{\overline{r},\overline{G}} - \sum_{t=1}^T r_t^\top q^{P,\pi_t}$, where:

$$\overline{r} := \begin{cases} \mathbb{E}_{r \sim \mathcal{R}}[r] & \text{with stochastic rewards} \\ \frac{1}{T}\sum_{t=1}^T r_t & \text{with adversarial rewards,} \end{cases} \quad \overline{G} := \begin{cases} \mathbb{E}_{G \sim \mathcal{G}}[G] & \text{with stochastic rewards} \\ \frac{1}{T}\sum_{t=1}^T G_t & \text{with adversarial rewards.} \end{cases}$$

Notice that the regret is computed with respect to the *optimal safe strategy in hindsight* in the adversarial case. We will refer to the optimal occupancy measure (the one associated with $\text{OPT}_{\overline{r},\overline{G}}$) as $q^*$, so that $\text{OPT}_{\overline{r},\overline{G}} = \overline{r}^\top q^*$ and the regret reduces to $R_T := \sum_{t=1}^T \overline{r}^\top (q^* - q^{P,\pi_t})$.

The cumulative constraint violation is defined as $V_T := \max_{i \in [m]} \sum_{t=1}^T \left[ G_t^\top q^{P,\pi_t} \right]_i$. For the sake of notation, we will refer to $q^{P,\pi_t}$ by using $q_t$, thus omitting the dependency on $P$ and $\pi$.

### 2.5 FEASIBILITY PARAMETER

We introduce a problem-specific parameter $\rho \in [0, L]$, which is strictly related to the feasibility of Problem (4). Formally, in settings with *stochastic constraints* chosen from a fixed distribution, the parameter $\rho$ is defined as $\rho := \max_{q \in \Delta(M)} \min_{i \in [m]} - \left[ \overline{G}^\top q \right]_i$. Instead, with *adversarial constraints*, the parameter $\rho$ is defined as $\rho := \max_{q \in \Delta(M)} \min_{t \in [T]} \min_{i \in [m]} - \left[ G_t^\top q \right]_i$. In both cases, the occupancy measure leading to the value of $\rho$ is denoted with $q^\circ$. Finally, we state the following condition on the value of $\rho$ which plays a central role when providing algorithm guarantees.

**Condition 2.** *It holds that $\rho \ge T^{-\frac{1}{8}} L \sqrt{20m}$.*

## 3  BEST-OF-BOTH-WORLDS CMDP OPTIMIZATION ALGORITHM

In this section, we present our algorithm named *primal-dual gradient descent online policy search* (PDGD-OPS). Its rationale is to instantiate two no-regret algorithms, referred to as *primal* and *dual player*, respectively. Precisely, the primal player optimizes on the primal variable space of the Lagrangian function, namely on the set $\Delta(M)$, while the dual player does it on the dual variable space $\mathbb{R}^m_{\geq 0}$, which, in our algorithm, is properly shrunk to $\left[0, T^{1/4}\right]^m$. As concerns the objective functions, the primal player aims to maximize the Lagrangian function, while the dual one to minimize it, as described in the Lagrangian 0-sum game defined in Corollary 1. Notice that, while the space of the dual variables is known *apriori*, the occupancy measure space needs be estimated online as the transition probabilities are unknown. Thus, it is necessary to employ a no-regret algorithm working with adversarial MDPs for the primal player. Moreover, in order to provide guarantees on the dynamics of the Lagrange multipliers—necessary to bound the cumulative regret and cumulative constraint violation—we require the primal player to satisfy the weak no-interval regret property (see the following Definition 3 for a formalization of such a property).

Algorithm 2 provides the pseudo-code of PDGD-OPS. As mentioned before, the algorithm employs two regret minimizers, named `UC-O-GDPS` and `OGD`, working on the space of the primal and dual variables, respectively. The occupancy measure is initialized uniformly (Line 1) by the primal player. We refer to Section 4 for the description of the `UC-O-GDPS` initialization. The dual player is initialized by the `OGD.INIT` procedure which takes as input the decision space $\mathcal{D}$ and a learning rate $\eta$, and it returns the vector $\lambda_1 = \underline{0}$ associated with the dual variable (Line 2).

During the learning process, at each episode $t \in [T]$, PDGD-OPS plays the policy $\pi^{\widehat{q}_t}$ induced by the occupancy measure $\widehat{q}_t$ computed in the previous episode (Line 5). The feedback received by the learner once the episode is concluded concerns the trajectory $(x_k, a_k)_{k=0}^{L-1}$ traversed in the CMDP, the reward vector, and the constraint matrix for that specific episode.

Given the observed feedback, the algorithm builds the Lagrangian objective function (Line 6) as the loss $\ell_t = G_t \lambda_t - r_t$ and feeds it to the primal player along with the trajectory and the adaptive learning rate (Line 7). The trajectory is needed to estimate the transition probabilities, while the rationale of the adaptive learning rate is to remove the quadratic dependence from $\|\lambda\|_1$ in the regret bound of the primal player. See Section 4 for the description of `UC-O-GDPS.UPDATE` (Line 8).

To conclude, we notice that the dual player receives only the loss $-G_t^\top \widehat{q}_t$, as, $r_t^\top \widehat{q}_t$ has no dependence on the optimization variable $\lambda_t$, thus, it does not affect the optimization process. For the sake of completeness, we report the `OGD` update of the dual player, namely `OGD.UPDATE` (Line 9) :

$$\lambda_{t+1} := \Pi_{\mathcal{D}} \left( \lambda_t + \eta G_t^\top \widehat{q}_t \right), \tag{5}$$

where $\Pi_{\mathcal{D}}$ is the Euclidean projection on the decision space $\mathcal{D}$, $\eta = \left[ K \sqrt{T \ln \left( \frac{T^2}{\delta} \right)} \right]^{-1}$ and $K$ is an instance-dependent quantity that does not depend on $T$ and $\delta$. From here on, we refer to the regret suffered by `OGD` with respect to a general Lagrange multiplier $\lambda$ as $R_T^{\mathsf{D}}(\lambda)$, where $\mathsf{D}$ stands for *dual*. Please notice that, thanks to the properties of `OGD` Orabona (2019) and using the aforementioned learning rate $\eta$, we obtain $R_T^{\mathsf{D}}(\lambda) \leq \tilde{\mathcal{O}} \left( (1 + \|\lambda\|_2^2) \sqrt{T} \right)$.

## 4  ADVERSARIAL MDP OPTIMIZATION ALGORITHM

We focus on the algorithm employed by the primal player. As aforementioned, this algorithm resorts to online learning techniques as the decision space of the primal player is not known beforehand. In particular, the algorithm is an online adversarial MDP optimizer, as Algorithm 2 deals with both stochastic and adversarial settings.

### 4.1  UC-O-GDPS ALGORITHM

*Upper confidence online gradient descent policy search* (UC-O-GDPS) follows the rationale of the UC-O-REPS algorithm by Rosenberg & Mansour (2019b), from which we highlight two major differences. The first difference concerns the update step. In particular, while in UC-O-REPS

the update is performed by online mirror descent when the unnormalized KL is used as Bregman divergence, in UC-O-GDPS such a step is performed by online gradient descent. The use of online gradient descent allows the UC-O-GDPS algorithm to satisfy the weak no-interval regret property (see Definition 3) which plays a central role in our regret analysis. We also notice that, to the best of our knowledge, the weak no-interval regret property has never been studied in episodic adversarial MDPs, and thus our result may be of independent interest. The second difference concerns the design of an adaptive learning rate which depends on the losses previously observed. The satisfaction of weak no-interval regret property and the adoption of our adaptive learning rate allow us to attain a regret bound of $\tilde{\mathcal{O}}(\sqrt{T})$ for PDGD-OPS in place of $\tilde{\mathcal{O}}\left(T^{3/4}\right)$.

**Transitions confidence set**   Initially, we discuss how UC-O-GDPS updates the confidence set, denoted with $\mathcal{P}$, on the transition probabilities $P$. This is done by following the approach prescribed by Rosenberg & Mansour (2019b). However, for the sake of clarity, we summarize the functioning. In particular, the update of the confidence set requires a non-negligible computational effort, however it is possible to update the confidence set at a subset of episodes to make the UC-O-GDPS algorithm more efficient without worsening the regret bounds. More precisely, the episodes are divided dynamically in epochs depending of the observed feedback, and the update of the confidence bound is only performed at the first episode of every epoch. UC-O-GDPS adopts counters of visits for each state-action pair $(x, a)$ and each state-action-state triple $(x, a, x')$ to estimate the empirical transition function as:

$$\overline{P}_i\left(x' \mid x, a\right) = \frac{M_i\left(x' \mid x, a\right)}{\max\left\{1, N_i(x, a)\right\}},$$

where $N_i(x, a)$ and $M_i\left(x' \mid x, a\right)$ are the initial values of the counters, that is, the total number of visits of pair $(x, a)$ and triple $(x, a, x')$, respectively, observed in the epochs preceding epoch $i$. Furthermore, a new epoch starts whenever there is a state-action pair whose counter is doubled compared to its initial value at the beginning of the epoch. The confidence set $\mathcal{P}_i$ is updated at every epoch $i$ as, for every $(x, a) \in X \times A$:

---

**Algorithm 2** PDGD-OPS

**Require:** $T, X, A, \delta$
1: $\widehat{q}_1 \leftarrow \texttt{UC-O-GDPS.INIT}\,(X, A, \delta)$
2: $\lambda_1 \leftarrow \texttt{OGD.INIT}\left(\left[0, T^{1/4}\right]^m, \eta\right)$
3: **for** $t = 1$ to $T$ **do**
4:     Play $\pi^{\widehat{q}_t}$ and observe trajectory $(x_k, a_k)_{k=0}^{L-1}$, reward vector $r_t$, and constraint matrix $G_t$
5:     
6:     $\ell_t \leftarrow G_t \lambda_t - r_t$
7:     $\eta_t = \frac{1}{\overline{\ell}_t C \sqrt{T}}$ with $\overline{\ell}_t = \max\{\|\ell_\tau\|_\infty\}_{\tau=1}^t$
8:     $\widehat{q}_{t+1} \leftarrow \texttt{UC-O-GDPS.UPDATE}\left(\ell_t, \eta_t, (x_k, a_k)_{k=0}^{L-1}\right)$
9:     $\lambda_{t+1} \leftarrow \texttt{OGD.UPDATE}\left(-G_t^\top \widehat{q}_t\right)$
10: **end for**

---

**Algorithm 3** `UC-O-GDPS.UPDATE`

**Require:** $\ell_t, \eta_t, (x_k, a_k)_{k=0}^{L-1}$
1: **for** $k \in [0..L-1]$ **do**
2:     Update counters:

$$N_i\left(x_k, a_k\right) \leftarrow N_i\left(x_k, a_k\right) + 1,$$
$$M_i\left(x_{k+1} \mid x_k, a_k\right) \leftarrow M_i\left(x_{k+1} \mid x_k, a_k\right) + 1$$

3: **end for**
4: **if** $\exists k, N_i\left(x_k, a_k\right) \geq \max\left\{1, 2N_{i-1}\left(x_k, a_k\right)\right\}$ **then**
5:     Increase epoch index $i \leftarrow i + 1$
6:     Initialize new counters: for all $(x, a, x')$,

$$N_i(x, a) = N_{i-1}(x, a)$$
$$M_i\left(x' \mid x, a\right) = M_{i-1}\left(x' \mid x, a\right)$$

7:     Update confidence set $\mathcal{P}_i$ as in Equation (6)
8: **end if**
9: Update occupancy measure:

$$\widehat{q}_{t+1} = \Pi_{\Delta(\mathcal{P}_i)}\left(\widehat{q}_t - \eta_t \ell_t\right)$$

---

$$\mathcal{P}_i = \left\{\widehat{P} : \left\|\widehat{P}\left(\cdot \mid x, a\right) - \overline{P}_i\left(\cdot \mid x, a\right)\right\|_1 \leq \epsilon_i\left(x, a\right) := \sqrt{\frac{2|X_{k(x)+1}|\ln\left(\frac{T|X||A|}{\delta}\right)}{\max\left\{1, N_i(x, a)\right\}}}\right\}, \quad (6)$$

where $k(x)$ denotes the index of the layer to which $x$ belongs and $\delta \in (0, 1)$ is the given confidence.

The next result, which directly follows from Rosenberg & Mansour (2019b), shows that the cumulative error due to the estimation of the transition probabilities grows sublinearly during time.

**Lemma 2.** *If the confidence set $\mathcal{P}$ is updated as in Equation (6), with probability at least $1 - 2\delta$ $\sum_{t=1}^{T} ||q_t - \widehat{q}_t||_1 \leq \mathcal{E}_\delta^q$, where $\mathcal{E}_\delta^q \leq \tilde{\mathcal{O}}(\sqrt{T})$.*

**Initialization**   UC-O-GDPS.INIT procedure (Line 1 of Algorithm 2) initializes the epoch index as $i = 1$ and confidence set $\mathcal{P}_1$ as the set of all possible transition functions. For all $k \in [0..L-1]$ and all $(x, a, x') \in X_k \times A \times X_{k+1}$, the counters are initialized as $N_0(x, a) = N_1(x, a) = M_0(x' \mid x, a) = M_1(x' \mid x, a) = 0$. Finally, the following occupancy measure

$$\widehat{q}_1(x, a, x') = \frac{1}{|X_k||A||X_{k+1}|}$$

is returned by the initialization procedure for all $k \in [0..L-1]$ and all $(x, a, x') \in X_k \times A \times X_{k+1}$.

**Update**   The pseudo-code of UC-O-GDPS.UPDATE procedure (used in Line 8 of Algorithm 2) is provided in Algorithm 3. Initially, it updates the estimate of the confidence set $\mathcal{P}$ (Lines 1–7) as described above, and, subsequently, it performs an update step according to projected online gradient descent (Line 9).

## 4.2   INTERVAL REGRET

Initially, we provide the definition of interval regret for adversarial online MDPs.

**Definition 2** (Interval regret). *Given an interval $[t_1..t_2] \subseteq [1..T]$, the interval regret with respect to a general occupancy measure $q$ is defined as:*

$$R_{t_1, t_2}(q) := \sum_{t=t_1}^{t_2} \ell_t^\top (q_t - q).$$

Now, we define the notion of weak no-interval regret. This notion plays a crucial role when proving the properties of Algorithm 2, and it is defined as follows.

**Definition 3** (Weak no-interval regret). *An online MDP optimizer satisfies the weak no-interval regret property if:*

$$R_{t_1, t_2}(q) \leq \tilde{\mathcal{O}}\left(\sqrt{T}\right) \quad \forall [t_1..t_2] \subseteq [1..T].$$

For the sake of clarity, in the following, we use the superscript $\mathsf{P}$ in the regret to distinguish the regret associated with the primal optimizer ($R^\mathsf{P}$) from the regret associated with the dual optimizer ($R^\mathsf{D}$), and we use $R_T^\mathsf{P}(q)$ in place of $R_{1,T}^\mathsf{P}(q)$. Next, we state the main result of this section.

**Theorem 3.** *With probability at least $1 - 2\delta$, when $\eta_t = \left(\bar{\ell}_t C \sqrt{T}\right)^{-1}$, UC-O-GDPS satisfies for any $q \in \cap_i \Delta(\mathcal{P}_i)$:*

$$R_{t_1, t_2}^\mathsf{P}(q) \leq \bar{\ell}_{t_1, t_2} \mathcal{E}_\delta^q + \bar{\ell}_{t_2} L C \sqrt{T} + \bar{\ell}_{t_2} \frac{|X||A|}{2} \frac{(t_2 - t_1 + 1)}{C \sqrt{T}},$$

*where $\bar{\ell}_{t_1, t_2} := \max\{||\ell_t||_\infty\}_{t=t_1}^{t_2}$, $\bar{\ell}_t := \bar{\ell}_{1,t}$ and $\delta \in [0, 1]$.*

Furthermore, it follows from Theorem 3 that, when $t_1 = 1, t_2 = T$, it holds $R_T^\mathsf{P} \leq \tilde{\mathcal{O}}\left(\bar{\ell}_T \sqrt{T}\right)$.

## 5   THEORETICAL RESULTS

In this section we provide the theoretical results attained by Algorithm 2 in terms of cumulative regret and cumulative constraint violation. We start providing a fundamental result on the Lagrange multiplier dynamics. Then, we distinguish two cases, which require different treatments. In the first, constraints are stochastic (Section 5.1), while in the second case they are adversarial (Section 5.2).

The main technical challenge when bounding the cumulative regret and constraint violation concerns bounding the space of the dual variables. We recall that, when employing standard no-regret

techniques, an unbounded dual space would lead to an unbounded loss for the primal regret minimizer, resulting in a linear regret. Our choice $\mathcal{D} = [0, T^{1/4}]^m$ of the dual decision space allows us to circumvent such an issue and PDGD-OPS to achieve a cumulative regret bound of $R_T \leq \tilde{\mathcal{O}}(T^{3/4})$, while keeping the cumulative violation sublinear. Nevertheless, when $\rho$ is large enough (namely, Condition 2 holds), the $\tilde{\mathcal{O}}\left(T^{3/4}\right)$ dependency in the upper bounds is not optimal. In particular, in this case, we can show that the Lagrangian vector never touches the boundaries of $\mathcal{D}$, and this property can be used to show that the regret and violation bounds are $\tilde{\mathcal{O}}(\sqrt{T})$. In the following, we present our result on how the Lagrange multipliers can be bounded, providing a proof sketch and referring to Appendix D for the complete proof.

**Theorem 4.** *If Condition 2 holds and PDGD-OPS is used, then, when $\zeta := \frac{20mL^2}{\rho^2}$, it holds*

$$||\lambda_t||_1 \leq \zeta \qquad \forall t \in [T+1]$$

*with probability at least $1 - 2\delta$ in the stochastic constraint setting and with probability at least $1 - \delta$ in the adversarial constraint setting.*

*Proof sketch.* The proof exploits the fact that both the primal and dual player satisfy the weak no-interval regret property. Precisely, the sum of the values of the Lagrangian function in $[t_1..t_2]$ can be lower bounded by using the interval regret of UC-O-GDPS, while the same quantity can be upper bounded with the interval regret of OGD, showing a contradiction concerning the value Lagrange multipliers can achieve for an opportune choice of constants and learning rates. $\square$

### 5.1 STOCHASTIC CONSTRAINT SETTING

The peculiarity of this setting is that, at every episode $t \in [T]$ the constraint matrix $G$ is sampled from a fixed distribution, namely $G_t \sim \mathcal{G}$. Instead, rewards $r_t$ can be sampled from a fixed distribution $\mathcal{R}$ or chosen adversarially.

**Azuma-Hoeffding bounds** Initially, we bound the error between the realizations of reward vectors and their corresponding mean values when the rewards are chosen stochastically. The proof is provided in Appendix D.

**Lemma 3.** *If the rewards are stochastic, then, with probability at least $1 - \delta$, it holds:*

$$\left| \sum_{t=1}^{T} (r_t - \bar{r})^\top q^* \right| \leq \mathcal{E}_\delta^r,$$

*where $\mathcal{E}_\delta^r := \frac{L}{\sqrt{2}}\sqrt{T \ln\left(\frac{2}{\delta}\right)}$.*

Now, we bound the error between the realizations of constraint violations and their corresponding mean values.

**Lemma 4.** *If the constraints are stochastic, given a sequence of occupancy measures $(q_t)_{t=1}^T$, then with probability at least $1 - \delta$, for all $[t_1..t_2] \subseteq [1..T]$, it holds:*

$$\left| \sum_{t=t_1}^{t_2} \lambda_t^\top \left( G_t^\top - \overline{G}^\top \right) q_t \right| \leq \lambda_{t_1,t_2} \mathcal{E}_{t_1,t_2,\delta}^G,$$

*where $\mathcal{E}_{t_1,t_2,\delta}^G := 2L\sqrt{2(t_2 - t_1 + 1)\ln\left(\frac{T^2}{\delta}\right)}$ and $\lambda_{t_1,t_2} := \max\{||\lambda_t||_1\}_{t=t_1}^{t_2}$.*

For the sake of notation, we use $\mathcal{E}_\delta^G$ in place of $\mathcal{E}_{1,T,\delta}^G$. Let us remark that $\mathcal{E}_\delta^r, \mathcal{E}_\delta^G \leq \tilde{\mathcal{O}}(\sqrt{T})$.

**Analysis when Condition 2 holds** We start by analyzing the case in which Condition 2 holds. By Theorem 4, we know that the maximum 1-norm of the dual vectors selected by OGD during the learning process is upper-bounded by the constant $\zeta$. Since $\zeta$ essentially determines the range of the Lagrangian function, we can prove optimal regret and violation bounds of order $\tilde{\mathcal{O}}\left(\zeta\sqrt{T}\right)$ for PDGD-OPS, as stated in the following theorem.

**Theorem 5.** *In the stochastic constraint setting, when Condition 2 holds, the cumulative regret and constraint violation incurred by PDGD-OPS are upper bounded as follows. If the rewards are adversarial, then with probability at least $1 - 4\delta$ Algorithm 2 provides $R_T \leq \zeta\mathcal{E}_\delta^G + \zeta\mathcal{E}_\delta^q + R_T^{\mathsf{D}}(\underline{0}) + R_T^{\mathsf{P}}(q^*)$ and $V_T \leq \frac{1}{\eta}\zeta + \mathcal{E}_\delta^q$. If the rewards are stochastic, then with probability at least $1 - 5\delta$ Algorithm 2 provides $R_T \leq \mathcal{E}_\delta^r + \zeta\mathcal{E}_\delta^G + \zeta\mathcal{E}_\delta^q + R_T^{\mathsf{D}}(\underline{0}) + R_T^{\mathsf{P}}(q^*)$, and $V_T \leq \frac{1}{\eta}\zeta + \mathcal{E}_\delta^q$. In both cases:*

$$R_T \leq \tilde{\mathcal{O}}\left(\zeta\sqrt{T}\right), \quad V_T \leq \tilde{\mathcal{O}}\left(\zeta\sqrt{T}\right).$$

Notice that, if Condition 2 does not hold, the bounds stated in Theorem 5 can become of order $\tilde{\mathcal{O}}\left(T^{3/4}\right)$ or even linear. We conclude the analysis of the stochastic constraint setting when Condition 2 holds with the following remark.

**Remark 1.** *In Theorem 5, the regret bound when the rewards are adversarial is better than the one when the rewards are chosen stochastically. This result may seem counter-intuitive as the adversarial setting is the hardest setting a learner might face. Informally, this is due to the different definition of the optimization baseline used in the stochastic and adversarial settings.*

**Analysis when Condition 2 does not hold.** We focus on the case in which Condition 2 does not hold. As previously observed, in this case the regret and violation bounds given in Theorem 5 are not meaningful anymore, as they could become linear in $T$ (in fact, this is exactly the case when $\rho \propto T^{-\frac{1}{4}}$). Nevertheless, by constraining the dual player to the decision space $\mathcal{D} = [0, T^{1/4}]^m$, we are able to prove worst-case regret and violation bounds of order $\tilde{\mathcal{O}}\left(T^{3/4}\right)$. This result is formalized in the following theorem.

**Theorem 6.** *In the stochastic constraint setting, when Condition 2 does not hold, the cumulative regret and constraint violations incurred by PDGD-OPS are upper bounded as follows. If the rewards are adversarial, then with probability at least $1 - 4\delta$ Algorithm 2 provides $R_T \leq mT^{\frac{1}{4}}\mathcal{E}_\delta^G + mT^{\frac{1}{4}}\mathcal{E}_\delta^q + R_T^{\mathsf{D}}(\underline{0}) + R_T^{\mathsf{P}}(q^*)$ and $V_T \leq (2 + 2L)\frac{1}{\eta}T^{\frac{1}{4}} + \mathcal{E}_\delta^q$. If the rewards are stochastic, then with probability at least $1 - 5\delta$ Algorithm 2 provides $R_T \leq \mathcal{E}_\delta^r + mT^{\frac{1}{4}}\mathcal{E}_\delta^G + mT^{\frac{1}{4}}\mathcal{E}_\delta^q + R_T^{\mathsf{D}}(\underline{0}) + R_T^{\mathsf{P}}(q^*)$ and $V_T \leq (2 + 2L)\frac{1}{\eta}T^{\frac{1}{4}} + \mathcal{E}_\delta^q$. In both cases, it holds:*

$$R_T \leq \tilde{\mathcal{O}}\left(T^{\frac{3}{4}}\right), \quad V_T \leq \tilde{\mathcal{O}}\left(T^{\frac{3}{4}}\right).$$

## 5.2 ADVERSARIAL CONSTRAINT SETTING

We recall that in this setting, at every episode $t \in [T]$, the constraint matrix $G_t$ is chosen adversarially. Instead, rewards $r_t$ can be sampled from a fixed distribution $\mathcal{R}$ or chosen adversarially. This case corresponds to the hardest scenario the learner can face. As stated in Section 2.5, the treatment of this case requires a definition of $\rho$ stronger than that used in the stochastic constraint setting. Thanks to such a redefinition, it is possible to achieve guarantees on the cumulative constraint violation of the same order of those attainable in the stochastic setting, while obtaining at least a constant fraction of the optimal reward. Such a result can be achieved when Condition 2 holds. Notice that both sublinear cumulative regret and sublinear cumulative constraint violation cannot be achieved in our setting, as shown by Mannor et al. (2009).

The following theorem summarizes our result for the adversarial constraint setting.

**Theorem 7.** *In the adversarial constraint setting, when Condition 2 holds, the cumulative regret and constraint violations incurred by PDGD-OPS are upper bounded as follows. If the rewards are adversarial, then with probability at least $1 - 2\delta$ Algorithm 2 provides $R_T \leq \frac{1}{1+\rho}T \cdot \mathrm{OPT}_{\overline{r},\overline{G}} + \zeta\mathcal{E}_\delta^q + R_T^{\mathsf{D}}(\underline{0}) + R_T^{\mathsf{P}}(\tilde{q})$ and $V_T \leq \frac{1}{\eta}\zeta + \mathcal{E}_\delta^q$. If the rewards are stochastic, then with probability at least $1 - 3\delta$ Algorithm 2 provides $R_T \leq \frac{1}{1+\rho}T \cdot \mathrm{OPT}_{\overline{r},\overline{G}} + \mathcal{E}_\delta^r + \zeta\mathcal{E}_\delta^q + R_T^{\mathsf{D}}(\underline{0}) + R_T^{\mathsf{P}}(\tilde{q})$ and $V_T \leq \frac{1}{\eta}\zeta + \mathcal{E}_\delta^q$. In both cases, it holds:*

$$\sum_{t=1}^{T} r_t^\top q_t \geq \Omega\left(\frac{\rho}{1+\rho}T \cdot \mathrm{OPT}_{\overline{r},\overline{G}}\right), \quad V_T \leq \tilde{\mathcal{O}}\left(\zeta\sqrt{T}\right).$$

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

# A    RELATED WORKS

In the following, we survey some previous works that are tightly related to ours. In particular, we first describe works dealing with the online learning problem in MDPs, and, then, we discuss some works studying the constrained version of the classical online learning problem.

**Online Learning in MDPs.**    There is a considerable literature on online learning problems (Cesa-Bianchi & Lugosi, 2006) in MDPs (see (Auer et al., 2008; Even-Dar et al., 2009; Neu et al., 2010) for some initial results on the topic). In such settings, two types of feedback are usually investigated: in the *full-information feedback* model, the entire loss function is observed after the learner's choice, while in the *bandit feedback* model, the learner only observes the loss due to the chosen action. Azar et al. (2017) study the problem of optimal exploration in episodic MDPs with unknown transitions and stochastic losses when the feedback is bandit. The authors present an algorithm whose regret upper bound is $\tilde{\mathcal{O}}(\sqrt{T})$, thus matching the lower bound for this class of MDPs and improving the previous result by Auer et al. (2008). Rosenberg & Mansour (2019b) study the online learning problem in episodic MDPs with adversarial losses and unknown transitions when the feedback is full information. The authors present an online algorithm exploiting entropic regularization and providing a regret upper bound of $\tilde{\mathcal{O}}(\sqrt{T})$. The same setting is investigated by Rosenberg & Mansour (2019a) when the feedback is bandit. In such a case, the authors provide a regret upper bound of the order of $\tilde{\mathcal{O}}(T^{3/4})$, which is improved by Jin et al. (2020) by providing an algorithm that achieves in the same setting a regret upper bound of $\tilde{\mathcal{O}}(\sqrt{T})$.

**Online Learning in CMDPs with Long-term Constraints.**    All the previous works on the topic study settings in which constraints are selected stochastically. In particular, Zheng & Ratliff (2020) deal with episodic CMDPs with stochastic losses and constraints, where the transition probabilities are known and the feedback is bandit. The regret upper bound of their algorithm is of the order of $\tilde{\mathcal{O}}(T^{3/4})$, while the cumulative constraint violation is guaranteed to be below a threshold with a given probability. Wei et al. (2018) deal with adversarial losses and stochastic constraints, assuming the transition probabilities are known and the feedback is full information. The authors present an algorithm that guarantees an upper bound of the order of $\tilde{\mathcal{O}}(\sqrt{T})$ on both regret and constraint violation. Bai et al. (2020) provide the first algorithm that achieves sublinear regret when the transition probabilities are unknown, assuming that the rewards are deterministic and the constraints are stochastic with a particular structure. Efroni et al. (2020) propose two approaches to deal with the exploration-exploitation dilemma in episodic CMDPs. These approaches guarantee sublinear regret and constraint violation when transition probabilities, rewards, and constraints are unknown and stochastic, while the feedback is bandit. Qiu et al. (2020) provide a primal-dual approach based on *optimism in the face of uncertainty*. This work shows the effectiveness of such an approach when dealing with episodic CMDPs with adversarial losses and stochastic constraints, achieving both sublinear regret and constraint violation with full-information feedback. Wei et al. (2023) and Ding & Lavaei (2023) consider the case in which rewards and constraints are non-stationary, assuming that their variation is bounded. Thus, their results are *not* applicable to general adversarial settings.

**Online Learning with Long-term Constraints.**    A central result is provided by Mannor et al. (2009), who show that it is impossible to suffer from sublinear regret and sublinear constraint violation when an adversary chooses losses and constraints. Liakopoulos et al. (2019) try to overcome such an impossibility result by defining a new notion of regret. They study a class of online learning problems with long-term budget constraints that can be chosen by an adversary. The learner's regret metric is modified by introducing the notion of a *K-benchmark*, *i.e.*, a comparator that meets the problem's allotted budget over any window of length $K$. Castiglioni et al. (2022a;b) deal with the problem of online learning with stochastic and adversarial losses, providing the first *best-of-both-worlds* algorithm for online learning problems with long-term constraints.

## B   EVENTS

Here we state the events that we use in the rest of the Appendix.

The following event states that the true occupancy measure space is always contained in the confidence set:

**Event $E^\Delta(\delta)$:**   $\Delta(M) \subseteq \cap_i \Delta(\mathcal{P}_i)$.

In particular, under $E^\Delta(\delta)$, we have that $q^\circ, q^* \in \cap_i \Delta(\mathcal{P}_i)$. $E^\Delta(\delta)$ holds with probability at least $1 - \delta$ (See Lemma 5).

The following event states that the cumulative error after $T$ episodes due to the difference between $q^{P,\pi_t}$ and $q^{P^{\widehat{q}_t},\pi_t}$ is small enough:

**Event**   $E^{\widehat{q}}(\delta)$**:**   $\sum_{t=1}^{T} ||q_t - \widehat{q}_t||_1 \leq \mathcal{E}_\delta^q$,   *where*   $\mathcal{E}_\delta^q := 4L|X|\sqrt{2T \ln\left(\frac{1}{\delta}\right)} + 6L|X|\sqrt{2T|A| \ln\left(\frac{T|X||A|}{\delta}\right)} \leq \tilde{\mathcal{O}}(\sqrt{T})$.

In the next sections we will often condition on the intersection of the previous events:

**Event $E^{\Delta,\widehat{q}}(\delta)$:**   $E^{\widehat{q}}(\delta) \cap E^\Delta(\delta)$

$E^{\Delta,\widehat{q}}(\delta)$ holds with probability at least $1 - 2\delta$ (See Lemma 2).

The next event states that, in case the rewards are stochastic, the reward accumulated is not too far from the mean reward accumulated.

**Event $E_{q^*}^r(\delta)$:**   $\left| \sum_{t=1}^{T} (r_t - \bar{r})^\top q^* \right| \leq \mathcal{E}_\delta^r$, *where* $\mathcal{E}_\delta^r = \frac{L}{\sqrt{2}} \sqrt{T \ln\left(\frac{2}{\delta}\right)} \leq \tilde{\mathcal{O}}\left(\sqrt{T}\right)$

$E_{q^*}^r(\delta)$ holds with probability at least $1 - \delta$ (See Lemma 3).

For the stochastic constraint setting, we define the quantity $\mathcal{E}_{t_1,t_2,\delta}^G := 2L\sqrt{2(t_2 - t_1 + 1)\ln\left(\frac{T^2}{\delta}\right)}$ and then two events bounding the cumulative difference between the dual utility with the average constraints and that with the sampled constraints.

**Event $E_{q^\circ}^G(\delta)$:**   *for all* $[t_1..t_2] \subseteq [1..T]$,   $\left| \sum_{t=t_1}^{t_2} \lambda_t^\top (G_t^\top - \overline{G}^\top) q^\circ \right| \leq \lambda_{t_1,t_2} \mathcal{E}_{t_1,t_2,\delta}^G$

**Event $E_{q^*}^G(\delta)$:**   *for all* $[t_1..t_2] \subseteq [1..T]$,   $\left| \sum_{t=t_1}^{t_2} \lambda_t^\top (G_t^\top - \overline{G}^\top) q^* \right| \leq \lambda_{t_1,t_2} \mathcal{E}_{t_1,t_2,\delta}^G$

$E_{q^\circ}^G(\delta), E_{q^*}^G(\delta)$ each hold with probability at least $1 - \delta$ (See Lemma 4). We denote $\mathcal{E}_\delta^G := \mathcal{E}_{1,T,\delta}^G$

# C  ADDITIONAL DETAILS AND OMITTED PROOF OF SECTION 4

## C.1  ALGORITHM

---

**Algorithm 4** Upper Confidence Online Gradient Descent Policy Search (UC-O-GDPS)

---

**Require:** state space $X$, action space $A$, episode number $T$, and confidence parameter $\delta$

1: Initialize epoch index $i = 1$ and confidence set $\mathcal{P}_1$ as the set of all transition functions. For all $k \in [0..L-1]$ and all $(x, a, x') \in X_k \times A \times X_{k+1}$, initialize counters $N_0(x, a) = N_1(x, a) = M_0(x' \mid x, a) = M_1(x' \mid x, a) = 0$ and occupancy measure

$$\widehat{q}_1(x, a, x') = \frac{1}{|X_k||A||X_{k+1}|}$$

  Initialize policy $\pi_1 = \pi^{\widehat{q}_1}$

2: **for** $t \in [T]$ **do**

3:   Execute policy $\pi_t$ for $L$ steps and obtain trajectory $x_k, a_k$ for $k \in [0..L-1]$ and loss $\ell_t$

4:   **for** $k \in [0..L-1]$ **do**

5:     Update counters:

$$N_i(x_k, a_k) \leftarrow N_i(x_k, a_k) + 1,$$
$$M_i(x_{k+1} \mid x_k, a_k) \leftarrow M_i(x_{k+1} \mid x_k, a_k) + 1$$

6:   **end for**

7:   **if** $\exists k, N_i(x_k, a_k) \geq \max\{1, 2N_{i-1}(x_k, a_k)\}$ **then**

8:     Increase epoch index $i \leftarrow i + 1$

9:     Initialize new counters: for all $(x, a, x')$,

$$N_i(x, a) = N_{i-1}(x, a)$$
$$M_i(x' \mid x, a) = M_{i-1}(x' \mid x, a)$$

10:     Update confidence set $\mathcal{P}_i$ based on Equation (6)

11:   **end if**

12:   Update occupancy measure:

13:   $\eta_t = \frac{1}{\overline{\ell}_t C \sqrt{T}}$ with $\overline{\ell}_t = \max\{||\ell_t||_\infty\}_{t=1}^{t}$

$$\widehat{q}_{t+1} = \Pi_{\Delta(\mathcal{P}_i)}(\widehat{q}_t - \eta_t \ell_t)$$

14:   Update policy $\pi_{t+1} = \pi^{\widehat{q}_{t+1}}$

15: **end for**

---

**Confidence Set.** The description of how Confidence Set on the Transition Probability functions are built and used, follows precisely the description of Rosenberg & Mansour (2019b). We report the functioning for completeness.

UC-O-GDPS keeps counters of visits of each state-action pair $(x, a)$ and each state-action-state triple $(x, a, x')$, in order to estimate the empirical transition function as:

$$\overline{P}_i(x' \mid x, a) = \frac{M_i(x' \mid x, a)}{\max\{1, N_i(x, a)\}}$$

where $N_i(x, a)$ and $M_i(x' \mid x, a)$ are the initial values of the counters, that is, the total number of visits of pair $(x, a)$ and triple $(x, a, x')$ respectively, before epoch $i$. Epochs are used to reduce the computational complexity; in particular, a new epoch starts whenever there exists a state-action whose counter is doubled compared to its initial value at the beginning of the epoch. Next, the confidence set for epoch $i$ is defined as:

$$\mathcal{P}_i = \left\{\widehat{P} : \left\|\widehat{P}(\cdot \mid x, a) - \overline{P}_i(\cdot \mid x, a)\right\|_1 \leq \epsilon_i(x, a) \quad \forall (x, a) \in X \times A\right\} \tag{7}$$

with $\epsilon_i(x, a)$ defined as:

$$\epsilon_i(x, a) = \sqrt{\frac{2|X_{k(x)+1}| \ln\left(\frac{T|X||A|}{\delta}\right)}{\max\{1, N_i(x, a)\}}}$$

using $k(x)$ for the index of the layer that $x$ belongs to and for some confidence parameter $\delta \in (0, 1)$. We state the following Lemma by Rosenberg & Mansour (2019b), which provides the results related to the confidence set $\epsilon_i(x, a)$.

**Lemma 5.** *Rosenberg & Mansour (2019b) For any $\delta \in [0, 1]$:*

$$\left\| P\left(\cdot | x, a\right) - \overline{P}_i\left(\cdot | x, a\right) \right\|_1 \leq \sqrt{\frac{2|X_{k(x)+1}| \ln\left(\frac{T|X||A|}{\delta}\right)}{\max\left\{1, N_i(x, a)\right\}}}$$

*holds with probability at least $1 - \delta$ simultaneously for all $(x, a) \in X \times A$ and all epochs.*

Lemma 5 implies that, with high probability, the occupancy measure space $\Delta(M)$ is included in the estimated one $\Delta(\mathcal{P}_i) \; \forall i$.

**Occupancy Measure Update.** The update of the occupancy measure is performed on the space $\Delta(\mathcal{P}_i)$, which is built on the estimated transition function set $\mathcal{P}_i$. More formally:

$$\widehat{q}_{t+1} = \Pi_{\Delta(\mathcal{P}_i)}\left(\widehat{q}_t - \eta_t \ell_t\right)$$

with $\eta_t = \frac{1}{\overline{\ell}_t C \sqrt{T}}$ with $\overline{\ell}_t = \max\{||\ell_t||_\infty\}_{t=1}^t$, and $C$ constant. The employment of Online Gradient Descent has been necessary to achieve the interval regret results, while the adaptive learning rate was chosen to improve the performance in terms of Regret bounds.

## C.2 Interval Regret

In the following subsections, we prove the theorem related to the interval regret of Algorithm 4. First, we will present the main theorem, then, all the necessary lemmas.

**Theorem 3.** *With probability at least $1 - 2\delta$, when $\eta_t = \left(\overline{\ell}_t C \sqrt{T}\right)^{-1}$, UC-O-GDPS satisfies for any $q \in \cap_i \Delta(\mathcal{P}_i)$:*

$$R^{\mathsf{P}}_{t_1, t_2}(q) \leq \overline{\ell}_{t_1, t_2} \mathcal{E}^q_\delta + \overline{\ell}_{t_2} L C \sqrt{T} + \overline{\ell}_{t_1, t_2} \frac{|X||A|}{2} \frac{(t_2 - t_1 + 1)}{C \sqrt{T}},$$

*where $\overline{\ell}_{t_1, t_2} := \max\{||\ell_t||_\infty\}_{t=t_1}^{t_2}$, $\overline{\ell}_t := \overline{\ell}_{1,t}$ and $\delta \in [0, 1]$.*

*Proof.* Assume Event $E^{\Delta, \widehat{q}}(\delta)$ holds. By definition 2:

$$R_{t_1, t_2}(q) = \sum_{t=t_1}^{t_2} \ell_t^\top (q_t - q)$$

$$= \underbrace{\sum_{t=t_1}^{t_2} \ell_t^\top (q_t - \widehat{q}_t)}_{①} + \underbrace{\sum_{t=t_1}^{t_2} \ell_t^\top (\widehat{q}_t - q)}_{②}$$

$$\leq \overline{\ell}_{t_1, t_2} \mathcal{E}^q_\delta + \overline{\ell}_{t_2} L C \sqrt{T} + \overline{\ell}_{t_1, t_2} \frac{|X||A|}{2} \frac{(t_2 - t_1 + 1)}{C \sqrt{T}}$$

where the Inequality holds by Lemmas 9 and 10. We focus on bounding the first term ① and the second term ②. $\qquad\square$

### C.2.1 Bound on the First Term

In order to bound the first term of the Interval Regret, we state some useful Lemmas by Rosenberg & Mansour (2019b).

**Lemma 6.** *Rosenberg & Mansour (2019b) Let $\{\pi_t\}_{t=1}^T$ be policies and let $\{P_t\}_{t=1}^T$ be transition functions. Then,*

$$\sum_{t=1}^T ||q^{P_t, \pi_t} - q^{P, \pi_t}||_1 \leq \sum_{t=1}^T \sum_{x \in X} \sum_{a \in A} |q^{P_t, \pi_t}(x, a) - q^{P, \pi_t}(x, a)| + \sum_{t=1}^T \sum_{x \in X} \sum_{a \in A} q^{P, \pi_t}(x, a) ||P_t(\cdot | x, a) - P(\cdot | x, a)||_1$$

(8)

*where $P_t = P^{\widehat{q}_t}$.*

The following Lemma, shows how to bound the first term in Equation (8) with the second one.

**Lemma 7.** *Rosenberg & Mansour (2019b) Let $\{\pi_t\}_{t=1}^T$ be policies and let $\{P_t\}_{t=1}^T$ be transition functions. Then, for every $k \in [1..L-1]$ and every $t = 1, ..., T$ it holds that:*

$$\sum_{x_k \in X_k} \sum_{a_k \in A} |q^{P_t, \pi_t}(x_k, a_k) - q^{P, \pi_t}(x_k, a_k)| \le \sum_{s=0}^{k-1} \sum_{x_s \in X_s} \sum_{a_s \in A} q^{P, \pi_t}(x_s, a_s) ||P_t(\cdot|x_s, a_s) - P(\cdot|x_s, a_s)||_1$$

*where $P_t = P^{\widehat{q_t}}$.*

and finally, Equation (8) is upper bounded given:

**Lemma 8.** *Rosenberg & Mansour (2019b) Let $\{\pi_t\}_{t=1}^T$ be policies and let $\{P_t\}_{t=1}^T$ be transition functions such that $q^{P_t, \pi_t} \in \Delta(\mathcal{P}_i)$ for every $t$. Then, with probability at least $1 - 2\delta$ Event $E^\Delta(\delta)$ holds and:*

$$\sum_{t=1}^T \sum_{k=0}^{L-1} \sum_{s=0}^{k-1} \sum_{x_s \in X_s} \sum_{a_s \in A} q^{P, \pi_t}(x_s, a_s) ||P_t(\cdot|x_s, a_s) - P(\cdot|x_s, a_s)||_1 \le 2L|X| \sqrt{2T \ln\left(\frac{1}{\delta}\right)} + 3L|X| \sqrt{2T|A| \ln\left(\frac{T|X||A|}{\delta}\right)}$$

*where $P_t = P^{\widehat{q_t}}$.*

From the previous Lemmas, it easy to show that:

**Lemma 2.** *If the confidence set $\mathcal{P}$ is updated as in Equation (6), with probability at least $1 - 2\delta$ $\sum_{t=1}^T ||q_t - \widehat{q_t}||_1 \le \mathcal{E}_\delta^q$, where $\mathcal{E}_\delta^q \le \tilde{\mathcal{O}}(\sqrt{T})$.*

*Proof.* Following Rosenberg & Mansour (2019b), by Lemmas 6, 7 and 8 we obtain that with probability at least $1 - 2\delta$ Event $E^\Delta(\delta)$ holds and: $\sum_{t=1}^T ||q^{P_t, \pi_t} - q^{P, \pi_t}||_1 \le 4L|X| \sqrt{2T \ln\left(\frac{1}{\delta}\right)} + 6L|X| \sqrt{2T|A| \ln\left(\frac{T|X||A|}{\delta}\right)}$ $\qquad \square$

Now, we are ready to bound ①.

**Lemma 9.** *Under Event $E^{\Delta, \widehat{q}}(\delta)$ it holds:*

$$\sum_{t=t_1}^{t_2} \ell_t^\top (q_t - \widehat{q_t}) \le \overline{\ell}_{t_1, t_2} \mathcal{E}_\delta^q$$

*with $\overline{\ell}_{t_1, t_2} := \max\{||\ell_t||_\infty\}_{t=t_1}^{t_2}$*

*Proof.*

$$\begin{aligned}
\sum_{t=t_1}^{t_2} \ell_t^\top (q_t - \widehat{q_t}) &\le \sum_{t=t_1}^{t_2} ||\ell_t||_\infty ||q_t - \widehat{q_t}||_1 \\
&\le \overline{\ell}_{t_1, t_2} \sum_{t=t_1}^{t_2} ||q_t - \widehat{q_t}||_1 \\
&\le \overline{\ell}_{t_1, t_2} \sum_{t=1}^{T} ||q_t - \widehat{q_t}||_1 \\
&\le \overline{\ell}_{t_1, t_2} \mathcal{E}_\delta^q \qquad\qquad (9)
\end{aligned}$$

*with $\overline{\ell}_{t_1, t_2} := \max\{||\ell_t||_\infty\}_{t=t_1}^{t_2}$ and where Inequality (9) holds under the event $E^{\widehat{q}}(\delta)$.* $\qquad \square$

### C.2.2 BOUND ON THE SECOND TERM

**Lemma 10.** *For any $q \in \cap_i \Delta(\mathcal{P}_i)$, the Projected OGD update:*

$$\widehat{q}_{t+1} = \Pi_{\Delta(\mathcal{P}_i)}(\widehat{q}_t - \eta_t \ell_t)$$

*with $\eta_t = \frac{1}{\overline{\ell}_t C \sqrt{T}}$ and $\overline{\ell}_t = \max\{||\ell_t||_\infty\}_{t=1}^t$ ensures:*

$$\sum_{t=t_1}^{t_2} \ell_t^\top (\widehat{q}_t - q) \le U_1 \frac{\overline{\ell}_{t_2}}{2} C\sqrt{T} + U_2 \frac{\overline{\ell}_{t_1,t_2}}{2} \frac{(t_2 - t_1 + 1)}{C\sqrt{T}}$$

*where $U_1 = 2L$, $U_2 = |X||A|$, $\overline{\ell}_{t_1,t_2} = \max\{||\ell_t||_\infty\}_{t=t_1}^{t_2}$.*

*Proof.* By the standard analysis of Projected Online Gradient Descent [Lemma 2.12 Orabona (2019)] we have:

$$\ell_t^\top (\widehat{q}_t - q) \le \frac{1}{2\eta_t} ||\widehat{q}_t - q||_2^2 - \frac{1}{2\eta_t} ||\widehat{q}_{t+1} - q||_2^2 + \frac{\eta_t}{2} ||\ell_t||_2^2.$$

Observe that for any two occupancy measures $q_1, q_2$ it holds:

$$
\begin{aligned}
||q_1 - q_2||_2^2 &\le ||q_1||_2^2 + ||q_2||_2^2 \\
&\le ||q_1||_1 + ||q_2||_1 \\
&\le 2L
\end{aligned}
$$

where the second Inequality follows from $q(x,a) \in [0,1] \;\; \forall x, a$. Then, summing over the interval $[t_1 .. t_2]$ we get:

$$
\begin{aligned}
\sum_{t=t_1}^{t_2} \ell_t^\top (\widehat{q}_t - q) \le & \frac{1}{2\eta_{t_1}} ||\widehat{q}_{t_1} - q||_2^2 \underbrace{- \frac{1}{2\eta_{t_2}} ||\widehat{q}_{t_2+1} - q||_2^2}_{\le 0} \\
& + \frac{1}{2} \sum_{t=t_1}^{t_2-1} \left( \frac{1}{\eta_{t+1}} - \frac{1}{\eta_t} \right) ||\widehat{q}_{t+1} - q||_2^2 + \frac{1}{2} \sum_{t=t_1}^{t_2} \eta_t ||\ell_t||_2^2 \\
\le & \frac{L}{\eta_{t_1}} + L \sum_{t=t_1}^{t_2-1} \left( \frac{1}{\eta_{t+1}} - \frac{1}{\eta_t} \right) + \frac{1}{2C\sqrt{T}} \sum_{t=t_1}^{t_2} \frac{1}{\overline{\ell}_t} \sum_{x,a} \ell_t(x,a)^2 && (10) \\
\le & \frac{L}{\eta_{t_1}} + L \underbrace{\sum_{t=t_1}^{t_2-1} \left( \frac{1}{\eta_{t+1}} - \frac{1}{\eta_t} \right)}_{= \frac{1}{\eta_{t_2}} - \frac{1}{\eta_{t_1}}} + \frac{1}{2C\sqrt{T}} \sum_{t=t_1}^{t_2} \underbrace{\frac{||\ell_t||_\infty}{\max\{||\ell_\tau||_\infty\}_{\tau=1}^t}}_{\le 1} ||\ell_t||_\infty \sum_{x,a} 1 \\
\le & L \overline{\ell}_{t_2} C\sqrt{T} + \frac{|X||A|}{2} \overline{\ell}_{t_1,t_2} \frac{(t_2 - t_1 + 1)}{C\sqrt{T}} && (11)
\end{aligned}
$$

where Inequality (10) follows from the definition of $\eta_t$, and from $\eta_t > \eta_{t+1}$, while Inequality (11) comes from the telescopic sum over $[t_1..t_2]$ and from the definition of $\eta_{t_2}$.

$\square$

## D   OMITTED PROOF OF SECTION 5

### D.1   INTERVAL REGRETS

In this section, we show the Interval Regrets, attained by both primal and dual player, in our specific framework.

### D.1.1 INTERVAL REGRET OF THE DUAL

In this subsection, we show the Interval Regret obtained by dual player. Recall that the dual variables are updated with Projected Online Gradient Descent as shown in (5) or equivalently:

$$\lambda_{t+1,i} = \min\left\{\max\left\{0, \lambda_{t,i} + \eta[G_t^\top]_i \widehat{q}_t\right\}, T^{1/4}\right\} \qquad (12)$$

with $\eta = \left[K\sqrt{T \ln\left(\frac{T^2}{\delta}\right)}\right]^{-1}$.

Let

$$R_{t_1,t_2}^{\mathsf{D}}(\lambda) := \sum_{t=t_1}^{t_2} (\lambda - \lambda_t)^\top G_t^\top \widehat{q}_t$$

denote the regret accumulated by OGD from episode $t_1$ to episode $t_2$ with respect to the constant multiplier $\lambda$. By standard analysis of OGD Orabona (2019) we have that:

$$R_{t_1,t_2}^{\mathsf{D}}(\lambda) \le \frac{||\lambda_{t_1} - \lambda||_2^2}{2\eta} + \frac{\eta}{2} \sum_{t=t_1}^{t_2} ||G_t^\top \widehat{q}_t||_2^2$$

We can upper-bound the quantity $||G_t^\top \widehat{q}_t||_2^2$ as:

$$||G_t^\top \widehat{q}_t||_2^2 = \sum_{i=1}^{m}\left(\sum_{x,a} g_{t,i}(x,a)\widehat{q}_t(x,a)\right)^2 \le \sum_{i=1}^{m}\left(\sum_{x,a} \widehat{q}_t(x,a)\right)^2 \le mL^2$$

obtaining:

$$R_{t_1,t_2}^{\mathsf{D}}(\lambda) \le D_1 \frac{||\lambda_{t_1} - \lambda||_2^2}{\eta} + D_2 \eta(t_2 - t_1 + 1)$$

with $D_1 = \frac{1}{2}$, $D_2 = \frac{mL^2}{2}$.

We bound the distance between lagrange multipliers for consecutive episodes.

**Lemma 11.** *If the dual player employs Projected Online Gradient Descent as in Update (12), it holds:*

$$||\lambda_{t+1}||_1 - ||\lambda_t||_1 \le m\eta L$$

*Proof.* Since the dual minimizer is performing projected gradient descent with learning rate $\eta$, and the gradient of the Lagrangian at time $t$ with respect to $\lambda$ is equal to $\widehat{q}_t^\top G_t^\top$, element-wise it holds that:

$$\begin{aligned}
\lambda_{t+1,i} &= \min\left\{\max\left\{0, \lambda_{t,i} + \eta[G_t^\top]_i \widehat{q}_t\right\}, T^{\frac{1}{4}}\right\} \\
&\le \max\left\{0, \lambda_{t,i} + \eta[G_t^\top]_i \widehat{q}_t\right\} \\
&\le \max\left\{0, \lambda_{t,i} + \eta||[G_t^\top]_i||_\infty ||\widehat{q}_t||_1\right\} \\
&\le \max\left\{0, \lambda_{t,i} + \eta L\right\} \\
&= \lambda_{t,i} + \eta L
\end{aligned}$$

Thus,

$$||\lambda_{t+1}||_1 - ||\lambda_t||_1 = \sum_{i=1}^{m}\lambda_{t+1,i} - \sum_{i=1}^{m}\lambda_{t,i} \le \sum_{i=1}^{m}\lambda_{t,i} + \sum_{i=1}^{m}\eta L - \sum_{i=1}^{m}\lambda_{t,i} = m\eta L$$

$\square$

### D.1.2 INTERVAL REGRET OF THE PRIMAL

We restate Lemma 10:

**Lemma 10.** *For any $q \in \cap_i \Delta(\mathcal{P}_i)$, the Projected OGD update:*

$$\widehat{q}_{t+1} = \Pi_{\Delta(\mathcal{P}_i)}(\widehat{q}_t - \eta_t \ell_t)$$

*with $\eta_t = \frac{1}{\overline{\ell}_t C \sqrt{T}}$ and $\overline{\ell}_t = \max\{||\ell_t||_\infty\}_{t=1}^t$ ensures:*

$$\sum_{t=t_1}^{t_2} \ell_t^\top (\widehat{q}_t - q) \leq U_1 \frac{\overline{\ell}_{t_2}}{2} C \sqrt{T} + U_2 \frac{\overline{\ell}_{t_1,t_2}}{2} \frac{(t_2 - t_1 + 1)}{C \sqrt{T}}$$

*where $U_1 = 2L$, $U_2 = |X||A|$, $\overline{\ell}_{t_1,t_2} = \max\{||\ell_t||_\infty\}_{t=t_1}^{t_2}$.*

Let

$$\lambda_{t_1,t_2} := \max\{||\lambda_t||_1\}_{t=t_1}^{t_2}.$$

Then it holds $\overline{\ell}_{t_1,t_2} \leq 1 + \lambda_{t_1,t_2}$ and we can restate the interval regret of the primal in terms of the 1-norm of the Lagrange multipliers as:

$$\sum_{t=t_1}^{t_2} r_t^{\mathcal{L}^\top} (q - \widehat{q}_t) \leq U_1 \frac{(1 + \lambda_{1,t_2})}{2} C \sqrt{T} + U_2 \frac{(1 + \lambda_{t_1,t_2})(t_2 - t_1 + 1)}{2} \frac{}{C \sqrt{T}}. \tag{13}$$

### D.2 BOUND ON THE LAGRANGE MULTIPLIERS

We prove Theorem 4, which we restate for convenience.

**Theorem 4.** *If Condition 2 holds and PDGD-OPS is used, then, when $\zeta := \frac{20mL^2}{\rho^2}$, it holds*

$$||\lambda_t||_1 \leq \zeta \qquad \forall t \in [T+1]$$

*with probability at least $1 - 2\delta$ in the stochastic constraint setting and with probability at least $1 - \delta$ in the adversarial constraint setting.*

*Proof.* Suppose event $E^\Delta(\delta)$ holds. If the constraints are stochastic, suppose event $E_{q^\circ}^G(\delta)$ holds too. Let $M > 1$ be a constant. We prove the statement by absurd. Suppose by absurd that there exists $t_2 \in [T]$ such that:

$$\forall t \leq t_2 \quad ||\lambda_t||_1 \leq \frac{2LM}{\rho^2} \qquad \wedge \qquad ||\lambda_{t_2+1}||_1 > \frac{2LM}{\rho^2}$$

and let $t_1 < t_2$ be such that:

$$||\lambda_{t_1-1}||_1 \leq \frac{2L}{\rho} \qquad \wedge \qquad \forall t : t_1 \leq t \leq t_2 \quad ||\lambda_t||_1 \geq \frac{2L}{\rho}.$$

By construction it holds that $1 < \frac{2L}{\rho} \leq ||\lambda_t||_1 \leq \frac{2LM}{\rho^2}$ for all $t_1 \leq t \leq t_2$. Also notice that by Lemma 11, for $\eta \leq \frac{1}{mL}$ it holds that:

$$||\lambda_{t_1}||_1 \leq ||\lambda_{t_1-1}||_1 + m\eta L \leq \frac{2L}{\rho} + m\eta L \leq \frac{4L}{\rho}$$

Focus on the quantity $\sum_{t=t_1}^{t_2} -\lambda_t^\top G_t^\top q^\circ$: in the stochastic constraint setting we have, under the event $E_{q^\circ}^G(\delta)$:

$$\sum_{t=t_1}^{t_2} -\lambda_t^\top G_t^\top q^\circ \geq \sum_{t=t_1}^{t_2} -\lambda_t^\top \overline{G}^\top q^\circ - \lambda_{t_1,t_2} \mathcal{E}_{t_1,t_2}^G$$

$$\geq \sum_{t=t_1}^{t_2} \sum_{i=1}^{m} -\lambda_{t,i} \left[\overline{G}^\top q^\circ\right]_i - \lambda_{t_1,t_2} \mathcal{E}_{t_1,t_2}^G$$

$$\geq \rho \sum_{t=t_1}^{t_2} \sum_{i=1}^{m} \lambda_{t,i} - \lambda_{t_1,t_2} \mathcal{E}_{t_1,t_2}^G$$

$$= \rho \sum_{t=t_1}^{t_2} ||\lambda_t||_1 - \lambda_{t_1,t_2} \mathcal{E}_{t_1,t_2}^G$$

$$\geq \rho \frac{2L}{\rho} (t_2 - t_1 + 1) - \lambda_{t_1,t_2} \mathcal{E}_{t_1,t_2}^G$$

$$= 2L(t_2 - t_1 + 1) - \lambda_{t_1,t_2} \mathcal{E}_{t_1,t_2}^G$$

While in the adversarial setting it holds:

$$\sum_{t=t_1}^{t_2} -\lambda_t^\top G_t^\top q^\circ \geq \sum_{t=t_1}^{t_2} \sum_{i=1}^{m} -\lambda_{t,i} \left[G_t^\top q^\circ\right]_i$$

$$\geq \rho \sum_{t=t_1}^{t_2} \sum_{i=1}^{m} \lambda_{t,i}$$

$$= \rho \sum_{t=t_1}^{t_2} ||\lambda_t||_1$$

$$\geq \rho \frac{2L}{\rho} (t_2 - t_1 + 1)$$

$$= 2L(t_2 - t_1 + 1)$$

In particular, we have that:

$$\sum_{t=t_1}^{t_2} -\lambda_t^\top G_t^\top q^\circ \geq 2L(t_2 - t_1 + 1) - \lambda_{t_1,t_2} \mathcal{E}_{t_1,t_2}^G$$

is true in both settings under the required events.

We can lower bound the cumulative value of the Lagrangian function, namely $r_t^{\mathcal{L}^\top} \widehat{q}_t$, from $t_1$ to $t_2$ by that achievable by the primal minimizer by always playing the feasible occupancy measure $q^\circ$:

$$\sum_{t=t_1}^{t_2} r_t^{\mathcal{L}^\top} \widehat{q}_t = \sum_{t=t_1}^{t_2} r_t^{\mathcal{L}^\top} q^\circ - \sum_{t=t_1}^{t_2} r_t^{\mathcal{L}^\top} (q^\circ - \widehat{q}_t)$$

$$= \underbrace{\sum_{t=t_1}^{t_2} r_t^\top q^\circ}_{\geq 0} + \sum_{t=t_1}^{t_2} -\lambda_t^\top G_t^\top q^\circ - \sum_{t=t_1}^{t_2} r_t^{\mathcal{L}^\top} (q^\circ - \widehat{q}_t)$$

$$\geq 2L(t_2 - t_1 + 1) - \lambda_{t_1,t_2} \mathcal{E}_{t_1,t_2,\delta}^G - \sum_{t=t_1}^{t_2} r_t^{\mathcal{L}^\top} (q^\circ - \widehat{q}_t)$$

Applying Lemma 10 and observing that by construction $1 \leq \lambda_{t_1,t_2} \leq \frac{2LM}{\rho^2}$, we can bound $1 + \lambda_{t_1,t_2} \leq \frac{4LM}{\rho^2}$ and obtain:

$$\sum_{t=t_1}^{t_2} r_t^{\mathcal{L}^\top} \widehat{q}_t \geq 2L(t_2 - t_1 + 1) - \frac{2LM}{\rho^2} \mathcal{E}_{t_1,t_2,\delta}^G - U_1 \frac{2LM}{\rho^2} C\sqrt{T} - U_2 \frac{2LM}{\rho^2} \frac{(t_2 - t_1 + 1)}{C\sqrt{T}}$$

since under $E^\Delta(\delta)$ we have that $q^\circ \in \cap_i \Delta(\mathcal{P}_i)$.

We can upper-bound the same quantity with the value achievable by the dual by always playing a vector of zeroes.

$$
\begin{aligned}
\sum_{t=t_1}^{t_2} r_t^{\mathcal{L}^\top} \widehat{q}_t &= \sum_{t=t_1}^{t_2} r_t^\top \widehat{q}_t - \sum_{t=t_1}^{t_2} \lambda_t^\top G_t^\top \widehat{q}_t \\
&\leq \sum_{t=t_1}^{t_2} r_t^\top \widehat{q}_t - \sum_{t=t_1}^{t_2} \underline{0}^\top G_t^\top \widehat{q}_t + R_{t_1,t_2}^{\mathsf{D}}(\underline{0}) \\
&\leq \sum_{t=t_1}^{t_2} L + D_1 \frac{\|\lambda_{t_1}\|_2^2}{\eta} + D_2 \eta(t_2 - t_1 + 1) \\
&\leq \sum_{t=t_1}^{t_2} L + D_1 \frac{\|\lambda_{t_1}\|_1^2}{\eta} + D_2 \eta(t_2 - t_1 + 1) \\
&\leq L(t_2 - t_1 + 1) + D_3 \frac{L^2}{\rho^2 \eta} + D_2 \eta(t_2 - t_1 + 1)
\end{aligned}
$$

With $D_3 = 4D_1$.

Combining the bounds on the cumulative value of the Lagrangian, we have:

$$2L(t_2 - t_1 + 1) - \frac{2LM}{\rho^2} \mathcal{E}_{t_1,t_2,\delta}^G - U_1 \frac{2LM}{\rho^2} C\sqrt{T} - U_2 \frac{2LM}{\rho^2} \frac{(t_2 - t_1 + 1)}{C\sqrt{T}}$$
$$\leq$$
$$L(t_2 - t_1 + 1) + D_3 \frac{L^2}{\rho^2 \eta} + D_2 \eta(t_2 - t_1 + 1)$$

Observing that $\mathcal{E}_{t_1,t_2,\delta}^G = 2L\sqrt{2(t_2 - t_1 + 1)\ln\left(\frac{T^2}{\delta}\right)} \leq U_3 l_1 \sqrt{t_2 - t_1 + 1}$ with $l_1 = \sqrt{\ln\left(\frac{T^2}{\delta}\right)}$ and $U_3 = 2L\sqrt{2}$ and rearranging the terms we obtain:

$$
\begin{aligned}
L(t_2 - t_1 + 1) \leq{}& U_3 \frac{2LM}{\rho^2} l_1 \sqrt{t_2 - t_1 + 1} + \\
&+ U_1 \frac{2LM}{\rho^2} C\sqrt{T} + \\
&+ U_2 \frac{2LM}{\rho^2} \frac{(t_2 - t_1 + 1)}{C\sqrt{T}} + \\
&+ D_2 \eta(t_2 - t_1 + 1) + \\
&+ D_3 \frac{1}{\eta} \frac{L^2}{\rho^2}
\end{aligned}
$$

We will make use of the following lemma:

**Lemma 12.** *For $\eta \leq \frac{1}{mL}$ and $\frac{M}{\rho} > 4$ it holds:*

$$(t_2 - t_1 + 1) > \frac{M}{\rho^2 m \eta}$$

*Proof.* By Lemma 11 we have:

$$\sum_{t=t_1}^{t_2} (||\lambda_{t+1}||_1 - ||\lambda_t||_1) \leq \sum_{t=t_1}^{t_2} m\eta L$$

which, since the sum in the LHS is telescopic, implies:

$$||\lambda_{t_2+1}||_1 - ||\lambda_{t_1}||_1 \leq (t_2 - t_1 + 1)m\eta L.$$

Also note that:

$$\frac{2LM}{\rho^2} - \frac{4L}{\rho} \leq ||\lambda_{t_2+1}||_1 - ||\lambda_{t_1}||_1.$$

Rearranging the terms, we obtain, for $\frac{M}{\rho} > 4$:

$$\frac{M}{\rho^2 m\eta} < \frac{2L(\frac{M}{\rho} - 2)}{\rho m\eta L} \leq (t_2 - t_1 + 1)$$

$\square$

Applying Lemma 12 we show that the above leads to a contradiction for some choices of $C$, $M$ and $\eta$, namely, we show that:

$$L(t_2 - t_1 + 1) > \quad U_3 \frac{2LM}{\rho^2} l_1 \sqrt{t_2 - t_1 + 1} + \tag{1}$$

$$+ U_1 \frac{2LM}{\rho^2} C\sqrt{T} + \tag{2}$$

$$+ U_2 \frac{2LM}{\rho^2} \frac{(t_2 - t_1 + 1)}{C\sqrt{T}} + \tag{3}$$

$$+ D_2 \eta (t_2 - t_1 + 1) + \tag{4}$$

$$+ D_3 \frac{1}{\eta} \frac{L^2}{\rho^2} \tag{5}$$

In the followings, we prove that each of the terms on the RHS is upper bounded by $\frac{1}{5}L(t_2 - t_1 + 1)$:

1. By trivial computations and applying Lemma 12:

$$\frac{1}{5}L(t_2 - t_1 + 1) > U_3 \frac{2LM}{\rho^2} l_1 \sqrt{T} \geq U_3 \frac{2LM}{\rho^2} l_1 \sqrt{t_2 - t_1 + 1}$$

$$(t_2 - t_1 + 1) > U_3 \frac{10M}{\rho^2} l_1 \sqrt{T}$$

$$(t_2 - t_1 + 1) > \frac{M}{\rho^2 m\eta} \geq U_3 \frac{10M}{\rho^2} l_1 \sqrt{T}$$

$$\frac{1}{m\eta} \geq 10U_3 l_1 \sqrt{T}$$

which is ensured by:

$$\boxed{\eta \leq \frac{1}{10mU_3 l_1 \sqrt{T}}}$$

2. Then applying again Lemma 12:

$$\frac{1}{5}L(t_2 - t_1 + 1) > U_1 \frac{2LM}{\rho^2} C\sqrt{T}$$

$$(t_2 - t_1 + 1) > \frac{M}{\rho^2 m\eta} \geq 10U_1 \frac{M}{\rho^2} C\sqrt{T}$$

which is true for:

$$\boxed{\eta \leq \frac{1}{10mU_1 C\sqrt{T}}}$$

3. We solve the third term with respect to $C$.

$$\frac{1}{5}L(t_2 - t_1 + 1) \geq U_2 \frac{2LM}{\rho^2} \frac{(t_2 - t_1 + 1)}{C\sqrt{T}}$$

which is ensured by:

$$\boxed{C \geq 10U_2 \frac{M}{\rho^2} \frac{1}{\sqrt{T}}}$$

4.

$$\frac{1}{5}L(t_2 - t_1 + 1) > D_2\eta(t_2 - t_1 + 1)$$

$$\frac{1}{5}L > D_2\eta$$

Which is ensured by

$$\boxed{\eta < \frac{L}{5D_2}}$$

5. Applying Lemma 12, we solve the Inequality with respect to M:

$$\frac{1}{5}L(t_2 - t_1 + 1) > D_3 \frac{1}{\eta}\frac{L^2}{\rho^2}$$

$$(t_2 - t_1 + 1) > \frac{M}{\rho^2 m\eta} \geq 5D_3 \frac{1}{\eta}\frac{L}{\rho^2}$$

$$\frac{M}{m} \geq 5D_3 L$$

from which:

$$\boxed{M \geq 5mD_3L}$$

We recall all the constants: $D_2 = \frac{mL^2}{2}$, $D_3 = 2$, $U_1 = 2L$, $U_2 = |X||A|$, $U_3 = 2L\sqrt{2}$. We choose $M = 10mL$ and recall Condition 2:

$$\rho \geq T^{-\frac{1}{8}}L\sqrt{20m} \quad \Rightarrow \quad \frac{20mL^2}{\rho^2} \leq T^{\frac{1}{4}} \leq \sqrt{T}$$

We now focus on the condition on $C$:

$$C \geq 10U_2 \frac{10mL}{\rho^2} \frac{1}{\sqrt{T}}$$

$$= 5\frac{U_2}{L}\frac{20mL^2}{\rho^2} \frac{1}{\sqrt{T}}$$

is thus always ensured by $C = 5\frac{U_2}{L}$. The conditions on $\eta$ are satisfied if:

$$\eta \leq \min\left\{ \frac{L}{5D_2}, \ \frac{1}{10mU_1C\sqrt{T}}, \ \frac{1}{10mU_3l_1\sqrt{T}} \right\}.$$

Observe that:

$$\min\left\{ \frac{L}{5D_2}, \ \frac{1}{10mU_1C\sqrt{T}}, \ \frac{1}{10mU_3l_1\sqrt{T}} \right\}$$

$$= \min\left\{ \frac{1}{2.5mL}, \ \frac{1}{10mU_1\left(\frac{5U_2}{L}\right)\sqrt{T}}, \ \frac{1}{20\sqrt{2}mLl_1\sqrt{T}} \right\}$$

which, if we plug in the value of $l_1$, leads to the choice:

$$\eta = \frac{1}{50m\max\left\{\frac{U_1U_2}{L}, \ L\right\}\sqrt{T\ln\left(\frac{T^2}{\delta}\right)}}$$

The remaining conditions $\frac{M}{\rho} > 4$, $\eta \leq \frac{1}{mL}$ are trivially satisfied. Summing the conditions $(1-5)$ proves the contradiction.

If we plug the values of $U_1$ and $U_2$ corresponding to UC-O-GDPS, we have $\max\left\{\frac{U_1 U_2}{L},\ L\right\} = \max\left\{2|X||A|,\ L\right\} = 2|X||A|$ and thus obtain:

$$\eta = \frac{1}{100m|X||A|\sqrt{T \ln\left(\frac{T^2}{\delta}\right)}}$$

$\square$

## D.3 ANALYSIS WITH STOCHASTIC CONSTRAINTS

### D.3.1 LOWER BOUND ON THE DUAL CUMULATIVE UTILITY

We start proving a useful Lemma in which we lower bound the dual cumulative utility. This Lemma holds both for the stochastic constraints and the adversarial constraint setting.

**Lemma 13.** *Under the event $E^{\widehat{q}}(\delta)$, the cumulative dual utility $\sum_{t=1}^{T} \lambda_t^\top G_t^\top q_t$ is lower bounded as:*

$$\sum_{t=1}^{T} \lambda_t^\top G_t^\top q_t \geq -\lambda_{1,T}\mathcal{E}_\delta^q - R_T^{\mathsf{D}}(\underline{0})$$

*where $\lambda_{t_1,t_2} := \max\{\|\lambda_t\|_1\}_{t=t_1}^{t_2}$.*

*Proof.* We exploit the fact that the dual is no-regret with respect to the $\underline{0}$ vector:

$$
\begin{aligned}
\sum_{t=1}^{T} \lambda_t^\top G_t^\top q_t &= \sum_{t=1}^{T} \lambda_t^\top G_t^\top (q_t - \widehat{q}_t) + \sum_{t=1}^{T} \lambda_t^\top G_t^\top \widehat{q}_t \\
&\geq \sum_{t=1}^{T} \lambda_t^\top G_t^\top (q_t - \widehat{q}_t) + \sum_{t=1}^{T} \underline{0}^\top G_t^\top \widehat{q}_t - R_T^{\mathsf{D}}(\underline{0}) \\
&\geq \sum_{t=1}^{T} - \underbrace{\|\lambda_t\|_1}_{\leq \lambda_{1,T}} \underbrace{\|G_t^\top\|_\infty}_{\leq 1} \|q_t - \widehat{q}_t\|_1 - R_T^{\mathsf{D}}(\underline{0}) \\
&\geq -\lambda_{1,T} \sum_{t=1}^{T} \|q_t - \widehat{q}_t\|_1 - R_T^{\mathsf{D}}(\underline{0}) \\
&\geq -\lambda_{1,T}\mathcal{E}_\delta^q - R_T^{\mathsf{D}}(\underline{0})
\end{aligned}
$$

where the last Inequality holds under $E^{\widehat{q}}(\delta)$. $\square$

### D.3.2 ANALYSIS WHEN CONDITION 2 HOLDS

We start by introducing the notation $\widehat{v}_{t,i} := [G_t^\top]_i \widehat{q}_t$, that is the violation of the $i$-th constraint incurred by $\widehat{q}_t$. We further denote $\widehat{V}_{t,i} := \sum_{\tau=1}^{t} \widehat{v}_{\tau,i}$. Observe that, when Condition 2 holds, thanks to Theorem 4 we have $\|\lambda_t\|_1 \leq T^{\frac{1}{4}}$ for all $t$ and thus $\lambda_{t,i} \leq T^{\frac{1}{4}}$. This means that $\lambda_{t,i}$ never gets past the upper extreme and the update of the dual is effectively equivalent to that of OGD working on the set $\mathrm{R}_{\geq 0}^m$:

$$\lambda_{t,i} = \max\{\lambda_{t,i} + \eta \widehat{v}_{t,i},\ 0\}$$

**Lemma 14.** *If Condition 2 holds, then for each episode $t \in [T]$ and each constraint $i$ it holds:*

$$\lambda_{t,i} \geq \eta \widehat{V}_{t-1,i}$$

*Proof.* We prove the result by induction. Suppose that the statement holds for episode $t$. Then

$$
\begin{aligned}
\lambda_{t+1,i} &= \max\{\lambda_{t,i} + \eta\widehat{v}_{t,i},\, 0\} \\
&\geq \lambda_{t,i} + \eta\widehat{v}_{t,i} \\
&\geq \eta\widehat{V}_{t-1,i} + \eta\widehat{v}_{t,i} \\
&= \eta\widehat{V}_{t,i}
\end{aligned}
$$

Observe that for $t = 1$ the statement holds as the sum on the RHS evaluates to 0. $\qquad\square$

**Lemma 15.** *If Condition 2 holds, under the events $E^\Delta(\delta)$, $E^{\widehat{q}}(\delta)$ and $E^G_{q^\circ}(\delta)$ for the stochastic constraint setting and under the events $E^\Delta(\delta)$ and $E^{\widehat{q}}(\delta)$ for the adversarial constraints one, it holds:*

$$
V_T \leq \widehat{V}_{T,i^*} + \mathcal{E}^q_\delta
$$

*Proof.* Let $i^*$ denote the most violated constraint, e.g. $i^* = \arg\max_i \sum_{t=1}^T [G_t^\top q_t]_i$. Then we have:

$$
\begin{aligned}
V_T &= \sum_{t=1}^T [G_t^\top q_t]_{i^*} \\
&= \sum_{t=1}^T [G_t^\top \widehat{q}_t]_{i^*} + \sum_{t=1}^T [G_t^\top (q_t - \widehat{q}_t)]_{i^*} \\
&= \widehat{V}_{T,i^*} + \sum_{t=1}^T [G_t^\top]_{i^*}(q_t - \widehat{q}_t) \\
&\leq \widehat{V}_{T,i^*} + \sum_{t=1}^T \|[G_t^\top]_{i^*}\|_\infty \|q_t - \widehat{q}_t\|_1 \\
&\leq \widehat{V}_{T,i^*} + \mathcal{E}^q_\delta
\end{aligned}
$$

Where the last step holds under $E^{\widehat{q}}(\delta)$ since $\|[G_t^\top]_{i^*}\|_\infty \leq 1$. $\qquad\square$

We are now ready to prove the regret and violation bounds for the stochastic constraint setting.

**Theorem 5.** *In the stochastic constraint setting, when Condition 2 holds, the cumulative regret and constraint violation incurred by PDGD-OPS are upper bounded as follows. If the rewards are adversarial, then with probability at least $1 - 4\delta$ Algorithm 2 provides $R_T \leq \zeta\mathcal{E}^G_\delta + \zeta\mathcal{E}^q_\delta + R^D_T(0) + R^P_T(q^*)$ and $V_T \leq \frac{1}{\eta}\zeta + \mathcal{E}^q_\delta$. If the rewards are stochastic, then with probability at least $1 - 5\delta$ Algorithm 2 provides $R_T \leq \mathcal{E}^r_\delta + \zeta\mathcal{E}^G_\delta + \zeta\mathcal{E}^q_\delta + R^D_T(0) + R^P_T(q^*)$, and $V_T \leq \frac{1}{\eta}\zeta + \mathcal{E}^q_\delta$. In both cases:*

$$
R_T \leq \tilde{\mathcal{O}}\left(\zeta\sqrt{T}\right), \quad V_T \leq \tilde{\mathcal{O}}\left(\zeta\sqrt{T}\right).
$$

*Proof.* Assume events $E^G_{q^\circ}(\delta)$, $E^G_{q^*}(\delta)$, $E^\Delta(\delta)$ and $E^{\widehat{q}}(\delta)$ hold.

Recall that $\lambda_{1,T} \leq \zeta$ under the events $E^\Delta(\delta)$ and $E^G_{q^\circ}(\delta)$ since Condition 2 holds (see proof of Theorem 4).

By Lemma 15 we have:

$$
\begin{aligned}
V_T &\leq \widehat{V}_{T,i^*} + \mathcal{E}^q_\delta \\
&\leq \frac{1}{\eta}\lambda_{T+1,i^*} + \mathcal{E}^q_\delta \\
&\leq \frac{1}{\eta}\|\lambda_{T+1}\|_1 + \mathcal{E}^q_\delta \\
&\leq \frac{1}{\eta}\zeta + \mathcal{E}^q_\delta
\end{aligned}
$$

Where the third Inequality holds for Lemma 14. By the definition of regret of the primal:

$$\sum_{t=1}^{T} r_t^\top q_t \geq \sum_{t=1}^{T} r_t^\top q^* - \sum_{t=1}^{T} \lambda_t^\top G_t^\top q^* + \sum_{t=1}^{T} \lambda_t^\top G_t^\top q_t - R_T^{\mathsf{P}}(q^*)$$

$$\geq \sum_{t=1}^{T} r_t^\top q^* - \sum_{t=1}^{T} \lambda_t^\top G_t^\top q^* - \lambda_{1,T} \mathcal{E}_\delta^q - R_T^{\mathsf{D}}(\underline{0}) - R_T^{\mathsf{P}}(q^*) \qquad (14)$$

$$\geq \sum_{t=1}^{T} r_t^\top q^* - \sum_{t=1}^{T} \lambda_t^\top \overline{G}^\top q^* - \lambda_{1,T} \mathcal{E}_\delta^G - \lambda_{1,T} \mathcal{E}_\delta^q - R_T^{\mathsf{D}}(\underline{0}) - R_T^{\mathsf{P}}(q^*) \qquad (15)$$

$$\geq \sum_{t=1}^{T} r_t^\top q^* - \sum_{t=1}^{T} \sum_{i} \lambda_{t,i} \underbrace{(\overline{G})_i q^*}_{\leq 0} - \lambda_{1,T} \mathcal{E}_\delta^G - \lambda_{1,T} \mathcal{E}_\delta^q - R_T^{\mathsf{D}}(\underline{0}) - R_T^{\mathsf{P}}(q^*) \qquad (16)$$

$$\geq \sum_{t=1}^{T} r_t^\top q^* - \zeta \mathcal{E}_\delta^G - \zeta \mathcal{E}_\delta^q - R_T^{\mathsf{D}}(\underline{0}) - R_T^{\mathsf{P}}(q^*)$$

where Inequality (14) holds for Lemma 13, and Inequality (15) holds under Event $E_{q^*}^G(\delta)$. We now focus on the case in which the rewards are adversarial. We have:

$$\sum_{t=1}^{T} r_t^\top q^* = T \cdot \overline{r}^\top q^* = T \cdot \mathrm{OPT}_{\overline{r}, \overline{G}}$$

and thus we obtain the stated bound:

$$\sum_{t=1}^{T} r_t^\top q_t \geq T \cdot \mathrm{OPT}_{\overline{r}, \overline{G}} - \zeta \mathcal{E}_\delta^G - \zeta \mathcal{E}_\delta^q - R_T^{\mathsf{D}}(\underline{0}) - R_T^{\mathsf{P}}(q^*)$$

By union bound on $E_{q^\circ}^G(\delta)$, $E_{q^*}^G(\delta)$ and $E^{\Delta, \widehat{q}}(\delta)$, the result holds with probability at least $1 - 4\delta$.

For the stochastic rewards case, we require also event $E_{q^*}^r(\delta)$ to hold. Thus,

$$\sum_{t=1}^{T} r_t^\top q^* \geq \sum_{t=1}^{T} \overline{r}^\top q^* - \mathcal{E}_\delta^r = T \cdot \mathrm{OPT}_{\overline{r}, \overline{G}} - \mathcal{E}_\delta^r$$

and thus we obtain the stated bound:

$$\sum_{t=1}^{T} r_t^\top q_t \geq T \cdot \mathrm{OPT}_{\overline{r}, \overline{G}} - \mathcal{E}_\delta^r - \zeta \mathcal{E}_\delta^G - \zeta \mathcal{E}_\delta^q - R_T^{\mathsf{D}}(\underline{0}) - R_T^{\mathsf{P}}(q^*)$$

By union bound on $E_{q^\circ}^G(\delta)$, $E_{q^*}^G(\delta)$, $E^{\Delta, \widehat{q}}(\delta)$ and $E_{q^*}^r(\delta)$, the result holds with probability at least $1 - 5\delta$.

Observe that under $E^{\Delta, \widehat{q}}(\delta)$ it holds:

$$R_T^{\mathsf{P}}(q^*) \leq \tilde{\mathcal{O}}\left( (1 + \lambda_{1,T}) \sqrt{T} \right) = \tilde{\mathcal{O}}\left( \zeta \sqrt{T} \right)$$

and

$$R_T^{\mathsf{D}}(\underline{0}) \leq \frac{mL^2}{2} \frac{1}{100 m |X| |A| \sqrt{\ln\left(\frac{T^2}{\delta}\right)}} \sqrt{T} \leq \mathcal{O}\left( \sqrt{T} \right)$$

$\square$

### D.3.3 ANALYSIS WHEN CONDITION 2 DOES NOT HOLD

**Lemma 16.** *If Condition 2 does not hold, then*

$$\widehat{V}_{T,i} \leq (2 + 2L)\frac{1}{\eta}T^{\frac{1}{4}} \qquad \forall T, i$$

*holds under the event $E^{\Delta}(\delta)$ in the adversarial constraint setting and under the events $E^{\Delta}(\delta)$, $E_{q^\circ}^{G}(\delta)$, in the stochastic constraint setting.*

*Proof.* Assume events $E^{\Delta}(\delta)$, $E_{q^\circ}^{G}(\delta)$ hold and suppose by absurd that $\widehat{V}_{T,i} = (2 + 2L + \epsilon)\frac{1}{\eta}T^{\frac{1}{4}}$, with $\epsilon > 0$, for some $T$ and $i$.

We can lower bound the quantity $\sum_{t=1}^{T} r_t^{\mathcal{L}\top}\widehat{q}_t$:

$$\sum_{t=1}^{T} r_t^{\mathcal{L}\top}\widehat{q}_t = \underbrace{\sum_{t=1}^{T} r_t^{\top} q^\circ}_{\geq 0} - \sum_{t=1}^{T} \lambda_t^{\top} G_t^{\top} q^\circ - \sum_{t=1}^{T} r_t^{\mathcal{L}\top}(q^\circ - \widehat{q}_t)$$

$$\geq \underbrace{-\sum_{t=1}^{T} \lambda_t^{\top} \overline{G}^{\top} q^\circ}_{\geq 0} - \lambda_{1,T}\mathcal{E}_{\delta}^{G} - \sum_{t=1}^{T} r_t^{\mathcal{L}\top}(q^\circ - \widehat{q}_t)$$

$$\geq -mT^{\frac{1}{4}}\mathcal{E}_{\delta}^{G} - \sum_{t=1}^{T} r_t^{\mathcal{L}\top}(q^\circ - \widehat{q}_t) \tag{17}$$

Where Inequality (17) holds since $||\lambda_t||_1 \leq mV^{\frac{1}{4}}$ by construction of the dual space. Observe that, if we are in the Adversarial setting, then from the (stronger) definition of $\rho$ and $q^\circ$ it holds $-\sum_{t=1}^{T} \lambda_t^{\top} G_t^{\top} q^\circ \geq 0$ and we obtain the tighter bound

$$\sum_{t=1}^{T} r_t^{\mathcal{L}\top}\widehat{q}_t \geq -\sum_{t=1}^{T} r_t^{\mathcal{L}\top}(q^\circ - \widehat{q}_t)$$

The dual is no regret with respect to the vector $\tilde{\lambda}$, whose elements are 0 for $j \neq i$ and $T^{\frac{1}{4}}$ in position $j = i$:

$$\sum_{t=1}^{T} r_t^{\mathcal{L}\top}\widehat{q}_t = \sum_{t=1}^{T} r_t^{\top}\widehat{q}_t - \sum_{t=1}^{T} \lambda_t^{\top} G_t^{\top}\widehat{q}_t$$

$$\leq \sum_{t=1}^{T} r_t^{\top}\widehat{q}_t - \sum_{t=1}^{T} \tilde{\lambda}^{\top} G_t^{\top}\widehat{q}_t + R_T^{\mathsf{D}}(\tilde{\lambda})$$

$$= \sum_{t=1}^{T} r_t^{\top}\widehat{q}_t - T^{\frac{1}{4}}\sum_{t=1}^{T} [G_t^{\top}\widehat{q}_t]_i + R_T^{\mathsf{D}}(\tilde{\lambda})$$

$$\leq LT - T^{\frac{1}{4}}\widehat{V}_{T,i} + R_T^{\mathsf{D}}(\tilde{\lambda})$$

Combining the bounds we have:

$$-mT^{\frac{1}{4}}\mathcal{E}_{\delta}^{G} - \sum_{t=1}^{T} r_t^{\mathcal{L}\top}(q^\circ - \widehat{q}_t) \leq LT - T^{\frac{1}{4}}\widehat{V}_{T,i} + R_T^{\mathsf{D}}(\tilde{\lambda})$$

$$T^{\frac{1}{4}}\widehat{V}_{T,i} \leq LT + mT^{\frac{1}{4}}\mathcal{E}_{\delta}^{G} + \sum_{t=1}^{T} r_t^{\mathcal{L}\top}(q^\circ - \widehat{q}_t) + R_T^{\mathsf{D}}(\tilde{\lambda})$$

$$\frac{\sqrt{T}}{\eta}(2 + 2L + \epsilon) \leq LT + mT^{\frac{1}{4}}\mathcal{E}_{\delta}^{G} + \sum_{t=1}^{T} r_t^{\mathcal{L}\top}(q^\circ - \widehat{q}_t) + R_T^{\mathsf{D}}(\tilde{\lambda}) \tag{18}$$

Observe that:

$$R_T^{\mathsf{D}}(\tilde{\lambda}) \leq \frac{1}{2}\frac{||\tilde{\lambda}||_2^2}{\eta} + \frac{mL^2}{2}\eta T = \frac{\sqrt{T}}{2\eta} + \frac{mL^2}{2}\frac{1}{100m|X||A|\sqrt{T\ln\left(\frac{T^2}{\delta}\right)}}T \leq L\frac{\sqrt{T}}{\eta}$$

Since $|X| \geq L$.

For the primal it holds by Lemma 10:

$$\sum_{t=1}^{T} r_t^{\mathcal{L}\top}(q^\circ - \hat{q}_t) = \sum_{t=1}^{T} \ell_t^\top(\hat{q}_t - q^\circ)$$

$$\leq \lambda_{1,T}U_1 C\sqrt{T} + \lambda_{1,T}U_2\frac{\sqrt{T}}{C}$$

$$\leq mT^{\frac{1}{4}}\sqrt{T}\left(U_1 C + \frac{U_2}{C}\right)$$

$$= m\left(U_1\frac{U_2}{5} + 5\right)\sqrt{T}\,T^{\frac{1}{4}}$$

$$= m\left(2L\frac{|X||A|}{5} + 5\right)\sqrt{T}\,T^{\frac{1}{4}}$$

$$\leq 6mL|X||A|\sqrt{T}\,T^{\frac{1}{4}}$$

$$\leq \frac{L}{\eta}T^{\frac{1}{4}} \leq L\frac{\sqrt{T}}{\eta}$$

And for the Azuma-Hoeffding term it holds:

$$mT^{\frac{1}{4}}\mathcal{E}_\delta^G = mT^{\frac{1}{4}}2L\sqrt{2T\ln\left(\frac{T^2}{\delta}\right)} \leq \frac{1}{\eta}T^{\frac{1}{4}} = \frac{\sqrt{T}}{\eta}$$

Observe that $LT \leq \frac{\sqrt{T}}{\eta}$ holds trivially.

Dividing both the terms in Equation (18) by $\frac{\sqrt{T}}{\eta}$, we obtain

$$2 + 2L + \epsilon \leq 2 + 2L$$

which is absurd. $\qquad\square$

We are now ready to prove the Regret and Violation bounds when Assumption 2 does not hold:

**Theorem 6.** *In the stochastic constraint setting, when Condition 2 does not hold, the cumulative regret and constraint violations incurred by PDGD-OPS are upper bounded as follows. If the rewards are adversarial, then with probability at least $1 - 4\delta$ Algorithm 2 provides $R_T \leq mT^{\frac{1}{4}}\mathcal{E}_\delta^G + mT^{\frac{1}{4}}\mathcal{E}_\delta^q + R_T^{\mathsf{D}}(\underline{0}) + R_T^{\mathsf{P}}(q^*)$ and $V_T \leq (2 + 2L)\frac{1}{\eta}T^{\frac{1}{4}} + \mathcal{E}_\delta^q$. If the rewards are stochastic, then with probability at least $1 - 5\delta$ Algorithm 2 provides $R_T \leq \mathcal{E}_\delta^r + mT^{\frac{1}{4}}\mathcal{E}_\delta^G + mT^{\frac{1}{4}}\mathcal{E}_\delta^q + R_T^{\mathsf{D}}(\underline{0}) + R_T^{\mathsf{P}}(q^*)$ and $V_T \leq (2 + 2L)\frac{1}{\eta}T^{\frac{1}{4}} + \mathcal{E}_\delta^q$. In both cases, it holds:*

$$R_T \leq \tilde{\mathcal{O}}\left(T^{\frac{3}{4}}\right), \quad V_T \leq \tilde{\mathcal{O}}\left(T^{\frac{3}{4}}\right).$$

*Proof.* Assume events $E^\Delta(\delta)$, $E^{\hat{q}}(\delta)$, $E_{q^*}^G(\delta)$, $E_{q^\circ}^G(\delta)$ hold. We avoid the computations and restart from (16), since the previous part of the proofs are identical:

$$\sum_{t=1}^{T} r_t^\top q_t \geq \sum_{t=1}^{T} r_t^\top q^* - \sum_{t=1}^{T}\sum_i \lambda_{t,i}\underbrace{(\overline{G})_i q^*}_{\leq 0} - \lambda_{1,T}\mathcal{E}_\delta^G - \lambda_{1,T}\mathcal{E}_\delta^q - R_T^{\mathsf{D}}(\underline{0}) - R_T^{\mathsf{P}}(q^*)$$

$$\geq \sum_{t=1}^{T} r_t^\top q^* - mT^{\frac{1}{4}}\mathcal{E}_\delta^G - mT^{\frac{1}{4}}\mathcal{E}_\delta^q - R_T^{\mathsf{D}}(\underline{0}) - R_T^{\mathsf{P}}(q^*)$$

By the same reasoning as in the proof of Theorem 5, we obtain that if the rewards are adversarial then

$$\sum_{t=1}^{T} r_t^\top q_t \geq T \cdot \text{OPT}_{\overline{r},\overline{G}} - mT^{\frac{1}{4}}\mathcal{E}_\delta^G - mT^{\frac{1}{4}}\mathcal{E}_\delta^q - R_T^D(\underline{0}) - R_T^P(q^*)$$

with probability at least $1 - 4\delta$ by union bound on $E^{\Delta,\widehat{q}}(\delta)$, $E_{q^*}^G(\delta)$ and $E_{q^\circ}^G(\delta)$, while if the rewards are stochastic, under the event $E_{q^*}^r(\delta)$ we have that:

$$\sum_{t=1}^{T} r_t^\top q_t \geq T \cdot \text{OPT}_{\overline{r},\overline{G}} - \mathcal{E}_\delta^r - mT^{\frac{1}{4}}\mathcal{E}_\delta^G - mT^{\frac{1}{4}}\mathcal{E}_\delta^q - R_T^D(\underline{0}) - R_T^P(q^*)$$

with probability at least $1 - 5\delta$ by union bound on $E^{\Delta,\widehat{q}}(\delta)$, $E_{q^*}^G(\delta)$, $E_{q^\circ}^G(\delta)$ and $E_{q^*}^r(\delta)$.

Observe that:

$$R_T^P(q^*) \leq \tilde{\mathcal{O}}\left(T^{\frac{3}{4}}\right)$$

and

$$R_T^D(\underline{0}) = \frac{mL^2}{2}\eta T \leq \tilde{\mathcal{O}}\left(\sqrt{T}\right).$$

In order to bound the violation, we apply Lemma 16:

$$V_T \leq \widehat{V}_{T,i^*} + \mathcal{E}_\delta^q \leq (2 + 2L)\frac{1}{\eta}T^{\frac{1}{4}} + \mathcal{E}_\delta^q$$

$\square$

## D.4 ANALYSIS WITH ADVERSARIAL CONSTRAINTS

### D.4.1 ANALYSIS WHEN CONDITION 2 HOLDS

**Theorem 7.** *In the adversarial constraint setting, when Condition 2 holds, the cumulative regret and constraint violations incurred by PDGD-OPS are upper bounded as follows. If the rewards are adversarial, then with probability at least $1 - 2\delta$ Algorithm 2 provides $R_T \leq \frac{1}{1+\rho}T \cdot \text{OPT}_{\overline{r},\overline{G}} + \zeta\mathcal{E}_\delta^q + R_T^D(\underline{0}) + R_T^P(\tilde{q})$ and $V_T \leq \frac{1}{\eta}\zeta + \mathcal{E}_\delta^q$. If the rewards are stochastic, then with probability at least $1 - 3\delta$ Algorithm 2 provides $R_T \leq \frac{1}{1+\rho}T \cdot \text{OPT}_{\overline{r},\overline{G}} + \mathcal{E}_\delta^r + \zeta\mathcal{E}_\delta^q + R_T^D(\underline{0}) + R_T^P(\tilde{q})$ and $V_T \leq \frac{1}{\eta}\zeta + \mathcal{E}_\delta^q$. In both cases, it holds:*

$$\sum_{t=1}^{T} r_t^\top q_t \geq \Omega\left(\frac{\rho}{1+\rho}T \cdot \text{OPT}_{\overline{r},\overline{G}}\right), \quad V_T \leq \tilde{\mathcal{O}}\left(\varsigma\sqrt{T}\right).$$

*Proof.* Assume events $E^\Delta(\delta)$ and $E^{\widehat{q}}(\delta)$ hold.

Recall that $\lambda_{1,T} \leq \zeta$ under the event $E^\Delta(\delta)$ since Condition 2 holds (see the proof of Theorem 4). Following the same steps of the proof of Theorem 5, we obtain:

$$V_T \leq \frac{1}{\eta}\zeta + \mathcal{E}_\delta^q$$

Let $\tilde{q} = \frac{\rho}{1+\rho}q^* + \frac{1}{1+\rho}q^\circ$, observe that it holds for all $t$ and for all $i$:

$$[G_t^\top \tilde{q}]_i = \frac{\rho}{1+\rho}\underbrace{[G_t^\top q^*]_i}_{\leq 1} + \frac{1}{1+\rho}\underbrace{[G_t^\top q^\circ]_i}_{\leq -\rho} \leq 0$$

$$r_t^\top \tilde{q} = \frac{\rho}{1+\rho}r_t^\top q^* + \frac{1}{1+\rho}r_t^\top q^\circ \geq \frac{\rho}{1+\rho}r_t^\top q^*$$

By the definition of regret of the primal:

$$\sum_{t=1}^{T} r_t^\top q_t \geq \sum_{t=1}^{T} r_t^\top \tilde{q} - \sum_{t=1}^{T} \lambda_t^\top G_t^\top \tilde{q} + \sum_{t=1}^{T} \lambda_t^\top G_t^\top q_t - R_T^{\mathsf{P}}(\tilde{q})$$

$$\geq \frac{\rho}{1+\rho} \sum_{t=1}^{T} r_t^\top q^* - \sum_{t=1}^{T} \sum_i \lambda_{t,i} \underbrace{[G_t^\top \tilde{q}]_i}_{\leq 0} + \sum_{t=1}^{T} \lambda_t^\top G_t^\top q_t - R_T^{\mathsf{P}}(\tilde{q})$$

$$\geq \frac{\rho}{1+\rho} \sum_{t=1}^{T} r_t^\top q^* - \lambda_{1,T} \mathcal{E}_\delta^q - R_T^{\mathsf{D}}(\underline{0}) - R_T^{\mathsf{P}}(\tilde{q})$$

$$\geq \frac{\rho}{1+\rho} \sum_{t=1}^{T} r_t^\top q^* - \zeta \mathcal{E}_\delta^q - R_T^{\mathsf{D}}(\underline{0}) - R_T^{\mathsf{P}}(\tilde{q})$$

where the third Inequality holds for Lemma 13.

By the same reasoning as in the proof of Theorem 5, we obtain that if the rewards are adversarial it holds:

$$\sum_{t=1}^{T} r_t^\top q_t \geq \frac{\rho}{1+\rho} T \cdot \mathrm{OPT}_{\overline{r},\overline{G}} - \zeta \mathcal{E}_\delta^q - R_T^{\mathsf{D}}(\underline{0}) - R_T^{\mathsf{P}}(\tilde{q})$$

$$= T \cdot \mathrm{OPT}_{\overline{r},\overline{G}} - \frac{1}{1+\rho} T \cdot \mathrm{OPT}_{\overline{r},\overline{G}} - \zeta \mathcal{E}_\delta^q - R_T^{\mathsf{D}}(\underline{0}) - R_T^{\mathsf{P}}(\tilde{q})$$

with probability at least $1 - 2\delta$, since we are conditioning on $E^{\Delta,\hat{q}}(\delta)$.

If the rewards are stochastic, requiring also event $E_{q^*}^r(\delta)$ to hold we obtain:

$$\frac{\rho}{1+\rho} \sum_{t=1}^{T} r_t^\top q^* \geq \frac{\rho}{1+\rho} \sum_{t=1}^{T} \overline{r}^\top q^* - \frac{\rho}{1+\rho} \mathcal{E}_\delta^r \geq \frac{\rho}{1+\rho} T \cdot \mathrm{OPT}_{\overline{r},\overline{G}} - \mathcal{E}_\delta^r$$

And thus,

$$\sum_{t=1}^{T} r_t^\top q_t \geq T \cdot \mathrm{OPT}_{\overline{r},\overline{G}} - \frac{1}{1+\rho} T \cdot \mathrm{OPT}_{\overline{r},\overline{G}} - \mathcal{E}_\delta^r - \zeta \mathcal{E}_\delta^q - R_T^{\mathsf{D}}(\underline{0}) - R_T^{\mathsf{P}}(\tilde{q})$$

with probability at least $1 - 3\delta$. Finally observe that, under Assumption 2 and event $E^{\Delta,\hat{q}}(\delta)$, it holds:

$$R_T^{\mathsf{P}}(\tilde{q}) \leq \tilde{\mathcal{O}}\left((1 + \lambda_{1,T})\sqrt{T}\right) \leq \tilde{\mathcal{O}}\left(\zeta\sqrt{T}\right)$$

and

$$R_T^{\mathsf{D}}(\underline{0}) \leq \frac{mL^2}{2} \frac{1}{100m|X||A|\sqrt{\ln\left(\frac{T^2}{\delta}\right)}} \sqrt{T} \leq \mathcal{O}\left(\sqrt{T}\right)$$

$\square$

## D.5 Azuma-Hoeffding Bounds and Proofs

In this subsection we prove that events $E_{q^*}^r(\delta)$, $E_{q^\circ}^G(\delta)$, $E_{q^*}^G(\delta)$ each hold with probability at least $1 - \delta$.

**Lemma 3.** *If the rewards are stochastic, then, with probability at least $1 - \delta$, it holds:*

$$\left|\sum_{t=1}^{T} (r_t - \overline{r})^\top q^*\right| \leq \mathcal{E}_\delta^r,$$

*where $\mathcal{E}_\delta^r := \frac{L}{\sqrt{2}} \sqrt{T \ln\left(\frac{2}{\delta}\right)}$.*

*Proof.* Observe that:

$$\max_{t \in [t_1..t_2]} \left| (r_t - \overline{r})^\top q^* \right| \leq \max_{t \in [t_1..t_2]} \underbrace{\|r_t - \overline{r}\|_\infty}_{\leq 1} \|q^*\|_1$$

$$\leq L$$

where the second Inequality holds since since $q^*(x,a) \geq 0$. By the Azuma-Hoeffding inequality for martingales we have that:

$$\mathbb{P}\left[ \left| \sum_{t=t_1}^{t_2} (r_t - \overline{r})^\top q^* \right| \geq \frac{L}{\sqrt{2}} \sqrt{T \ln\left(\frac{2}{\delta}\right)} \right] \leq \delta.$$

$\square$

We perform the same analysis for the constraints, obtaining:

**Lemma 4.** *If the constraints are stochastic, given a sequence of occupancy measures* $(q_t)_{t=1}^T$, *then with probability at least* $1 - \delta$, *for all* $[t_1..t_2] \subseteq [1..T]$, *it holds:*

$$\left| \sum_{t=t_1}^{t_2} \lambda_t^\top \left( G_t^\top - \overline{G}^\top \right) q_t \right| \leq \lambda_{t_1,t_2} \mathcal{E}_{t_1,t_2,\delta}^G,$$

*where* $\mathcal{E}_{t_1,t_2,\delta}^G := 2L\sqrt{2(t_2 - t_1 + 1)\ln\left(\frac{T^2}{\delta}\right)}$ *and* $\lambda_{t_1,t_2} := \max\{\|\lambda_t\|_1\}_{t=t_1}^{t_2}$.

*Proof.* Observe that:

$$\max_{t \in [t_1..t_2]} \left| \lambda_t^\top (G_t^\top - \overline{G}^\top) q_t \right| \leq \max_{t \in [t_1..t_2]} \|\lambda_t\|_1 \underbrace{\left\| G_t^\top - \overline{G}^\top \right\|_\infty}_{\leq 2} \|q_t\|_1$$

$$\leq \max_{t \in [t_1..t_2]} 2\|\lambda_t\|_1 L$$

$$= 2\lambda_{t_1,t_2} L$$

where the second Inequality holds since $q_t(x,a) \geq 0$ and $\lambda_{t,i} \geq 0$. By the Azuma-Hoeffding inequality for martingales we have that:

$$\mathbb{P}\left[ \left| \sum_{t=t_1}^{t_2} \lambda_t^\top (G_t^\top - \overline{G}^\top) q_t \right| \geq 2\lambda_{t_1,t_2} L \sqrt{2(t_2 - t_1 + 1)\ln\left(\frac{2}{\delta}\frac{T^2}{2}\right)} \right] \leq 2\delta/T^2.$$

A union bound over all the $t_1, t_2$ such that $[t_1..t_2] \subseteq [1..T]$ concludes the proof. $\square$

