# OpenReview forum: "A Best-of-Both-Worlds Algorithm for MDPs with Long-Term Constraints"
_ICLR.cc/2024/Conference — Submitted to ICLR 2024_

### Official Review · Reviewer_Txce · 2023-10-19

**Soundness:** 3 good
**Presentation:** 3 good
**Contribution:** 2 fair
**Rating:** 5
**Confidence:** 3

**Summary:**

This paper studies online learning in episodic constrained markov decision processes. They study the most general setting, where both the rewards and the constraints are chosen either stochastically or adversarially, and the transition is unknown to the learner. They provide the first best-of-both-worlds algorithm, where they can achieve optimal regret and constraint violation (in terms of the number of episodes) when constraints are selected stochastically, and provide the first guarantee when the constraints are chosen adversarially.

**Strengths:**

The paper is well written and easy to understand. The proposed algorithm is simple to implement and yet quite general and widely applicable. Moreover, their algorithm can achieve optimal regret bound and constraint violation (in terms of the number of episodes) when the constraints are chosen stochastically, which is the most common setting.

**Weaknesses:**

Both the algorithms and results in (Castiglioni et al., 2022b) and this paper look similar and the only difference I see is that this paper extends to MDP. So I am doubtful about the technical contributions of this paper. I would suggest the author highlighting the novelty of this work.

**Questions:**

1. What are the main technical contributions of this paper (other than extending (Castiglioni et al., 2022b) to MDP)?
2. Can your results be generalized to discounted MDP or Stochastic Shortest Path easily?

---

> ### Author Response · Authors · 2023-11-15
>
> > Both the algorithms and results in (Castiglioni et al., 2022b) and this paper look similar and the only difference I see is that this paper extends to MDP. So I am doubtful about the technical contributions of this paper. I would suggest the author highlighting the novelty of this work. What are the main technical contributions of this paper (other than extending (Castiglioni et al., 2022b) to MDP)?
>
> We thank the Reviewer for the opportunity to clarify these fundamental aspects. We will surely strengthen the original contribution section in the final version of the paper including the following discussion.
>
> We believe that our work provide quite novel techniques with respect to (Castiglioni et al., 2022b). Castiglioni et al. focus on general constraints but they assume to know the feasibility parameter $\rho$. Please notice that, assuming the knowledge of $\rho$ is arguably an unreasonable assumption in real-world scenarios, since it almost never happens to know a-priori the performances, in terms of violation, of the strictly feasible solution. Castiglioni et al. were able to relax this assumption only in the stochastic setting, while in our paper we relax this assumption in both the adversarial and the stochastic setting.
> This is a non-trivial task and indeed this assumption is present in all subsequent works on general constraints (see, e.g., ["Contextual Bandits with Packing and Covering Constraints: A Modular Lagrangian Approach via Regression.", 2023]).
>
> Precisely, even if our work is related to (Castiglioni et al., 2022b), the following technical contributions of our paper are novel with respect to the contributions in (Castiglioni et al., 2022b):
>
> i) We show how to ensure that the Lagrange multipliers are kept small without any explicit constraint and any knowledge of the value of the strictly feasibility parameter $\rho$. To do so, we are the first to employ "no-interval regret" properties for the primal and dual regret minimizers. The idea of using no-interval regret minimizers allows us to prove that the Lagrange multipliers are automatically bounded during the learning dynamic, while previous works with adversarial constraints need to bound the Lagrange multipliers a priori.
>
> ii) We design the first adaptive regret minimizer for adversarial MDPs. State-of-the-art MDPs online learning techniques rely on OMD-like (with KL regularizer) optimization updates, which make the no-interval regret property unattainable. Resorting to OGD-like updates and changing the theoretical analysis allows us to build an algorithm which is no-regret for every interval of the learning dynamic, thus having more guarantees in terms of stability.
> The aforementioned achievement may push the online learning research area to develop more stable regret minimizer even in complex environments such as Markov decision processes, since the classical static regret definition might not always make sense, especially when the environment is non-stationary such that there is no single fixed action that performs well in every interval of the horizon.
>
> To summarize, even if some results seem to be in common with (Castiglioni et al., 2022b), the theoretical analysis are quite independent. Indeed, as previously specified, the main theoretical effort of our work lies in showing the bound on the Lagrangian multiplier given the no-interval regret property of both the primal and the dual regret minimizers. These kind of analysis cannot be found in (Castiglioni et al., 2022b).
>
> >Can your results be generalized to discounted MDP or Stochastic Shortest Path easily?
>
> Our setting without constraints is sometimes referred to as the stochastic shortest path problem in the literature. Thus, for the stochastic shortest path problem, there is no need of generalization. On the contrary, our techniques cannot be generalized to infinite horizon (thus, discounted) MDPs. Nevertheless, to the best of our knowledge, there are not works on online learning in adversarial MDPs with infinite horizon (when the transitions are not known), even in the simpler unconstrained setting.

---

> > ### Comment · Reviewer_Txce · 2023-11-16
> > **Reply to rebuttal**
> >
> > > Our setting without constraints is sometimes referred to as the stochastic shortest path problem in the literature.
> >
> > The setting without constraint is a finite-horizon problem, not the general stochastic shortest path problem. In a stochastic shortest path model, the horizon is varying depending on when the agent reaches the goal state. See [1].
> >
> > [1] Dimitri P Bertsekas and John N Tsitsiklis. An analysis of stochastic shortest path problems. Mathematics of Operations Research, 16(3):580–595, 1991.

---

> > > ### Author Response · Authors · 2023-11-16
> > >
> > > > The setting without constraint is a finite-horizon problem, not the general stochastic shortest path problem. In a stochastic shortest path model, the horizon is varying depending on when the agent reaches the goal state. See [1].
> > > >
> > > > [1] Dimitri P Bertsekas and John N Tsitsiklis. An analysis of stochastic shortest path problems. Mathematics of Operations Research, 16(3):580–595, 1991.
> > >
> > > We are sorry for the misunderstanding, but, in the online learning literature, there are multiple definitions of the stochastic shortest path problem. As the Reviewer noticed, our algorithm cannot handle the general stochastic shortest path problem as defined in [1]. Indeed, our framework cannot handle possibly infinite-horizon problems. Nevertheless, there is a rich literature on finite-horizon stochastic shortest path problems. In such a setting, our algorithm works. Precisely, we referred to the following works in which our setting  (without constraints) is sometimes called the online stochastic shortest path problem, namely ["Online Stochastic Shortest Path with Bandit Feedback and Unknown Transition Function", 2019] and ["The Online Loop-free Stochastic Shortest-Path Problem", 2010]. Indeed, if we focus on the stochastic shortest path problem with the loop-free assumption (which is standard in the online adversarial MDPs literature), the problem is actually a loop-free MPD in which the target state is the last one (see, e.g., ["The Online Loop-free Stochastic Shortest-Path Problem", 2010], ["Online Stochastic Shortest Path with Bandit Feedback and Unknown Transition Function", 2019], ["Online Convex Optimization in Adversarial Markov Decision Processes", 2019], ["Learning Adversarial MDPs with Bandit Feedback and Unknown Transition", 2020], ["Policy Optimization in Adversarial MDPs: Improved Exploration via Dilated Bonuses", 2021]).
> > >
> > > We hope that we have at least partially addresses the Reviewer's concerns, and we hope that the Reviewer is willing to reconsider their evaluation of our paper.

---

### Official Review · Reviewer_KHRX · 2023-10-29

**Soundness:** 3 good
**Presentation:** 3 good
**Contribution:** 2 fair
**Rating:** 3
**Confidence:** 5

**Summary:**

The paper studies the online CMDP problem where the rewards and constraints can be selected either stochastically or adversarially. The authors propose a best-of-both-worlds algorithm for both settings. For the stochastic setting, the result matches the existing known result when a similar slater condition holds. When the condition doesn’t hold, all the bounds become $O(T^{3/4}).$ For the adversarial constraint setting, a competitive ratio result is provided, while the violation is $O(\zeta\sqrt{T}).$

**Strengths:**

The paper studies online learning in CMDPs with adversarial constraints. The proposed approach is well-presented, and the paper is easy to follow. The algorithm does not require knowledge of the Slater-like condition, and the theoretical results depend on the upper bound of the Lagrangian space, indicating the problem's difficulty under varying conditions.

**Weaknesses:**

- The technical contributions appear somewhat limited, as the algorithm follows a standard framework in CMDPs, and its dependence on $\rho$ is highly motivated by prior work [1].

- The term 'best-of-both worlds' is somewhat misleading. In the context of bandits and RL, 'best-of-both-worlds' typically implies achieving $\sqrt{T}$ regret in the adversarial setting versus $\log{T}$ regret in the stochastic setting. This paper only demonstrates a $\sqrt{T}$ rate in a fully stochastic setting when condition 2 holds. Furthermore, the 'best-of-both-worlds' literature in CMDPs is not discussed at all. Additionally, many algorithms can achieve zero constraint violations. Therefore it's unclear if this is the best achievable result.

- All the theoretical results are standard, and the use of projection to bound the dual variable for boundedness is not surprising.

- In the unconstrained case, several works have discussed bandit feedback and constraint violation without cancellation. However, the authors have not provided much insight into the challenges of removing these assumptions and studying the stronger setting.

[1] Matteo Castiglioni, Andrea Celli, Alberto Marchesi, Giulia Romano, and Nicola Gatti. A unifying framework for online optimization with long-term constraints.

**Questions:**

The regret bounds depend on condition 2. How is this related to the Slater condition, and which one is stronger? If they are not closely related, what happens if condition 2 doesn't hold but the Slater condition does? In such a case, your algorithm cannot achieve results of the same order as an algorithm using the Slater condition. How can we even say if it is the best of both worlds?

---

> ### Author Response · Authors · 2023-11-15
>
> >The technical contributions appear somewhat limited, as the algorithm follows a standard framework in CMDPs, and its dependence on $\rho$ is highly motivated by prior work [1].
>
> We believe that the Reviewer is missing some crucial aspects of our technical contributions. Let us remark that our paper is the first one in the literature to address the problem with adversarial constraints in CMDPs, which is the hardest possible setting. Moreover, our paper does so by proposing a novel best-of-both-world algorithm that is able to seamlessly handle adversarial/stochastic rewards and constraints, and this algorithm is the first of its kind in CMDPs. Our algorithm is based on a primal-dual framework that is fundamentally different from the techniques usually employed in the literature on CMDPs, as having to deal with adversarial constraints prevents us from using any kind of confidence bounds, which are instead generally used in the literature. Moreover, even when primal-dual techniques are employed in place of techniques based on confidence bounds, the analysis has never been extended to adversarial constraints.
>
> As concerns the differences of our work with respect to [1], in the following we detail the several aspects in which the two works depart. Crucially, [1] focuses on general constraints but they assume to know the feasibility parameter $\rho$. Then, it is possible to upper bound the Lagrange multipliers with $\frac{1}{\rho}$. Please notice that, assuming the knowledge of $\rho$ is arguably an unreasonable assumption in real-world scenarios, since it almost never happens to know a priori the performances, in terms of violation, of the strictly feasible solution. Castiglioni et al. were able to relax this assumption only in the stochastic setting, while in our paper we relax this assumption in both the adversarial and the stochastic setting.
>
> Precisely, the following technical contributions of our paper are novel with respect to the contributions in [1]:
>
> i) We show how to ensure that the Lagrange multipliers are kept small without any explicit constraint and any knowledge of the value of the strict feasibility parameter $\rho$. To do so, we are the first to employ no-interval regret properties for the primal and dual regret minimizers. The idea of using no-interval regret minimizers allows us to prove that the Lagrange multipliers are automatically bounded during the learning dynamic, while previous works with adversarial constraints need to bound the Lagrange multipliers a priori.
>
> ii) We design the first adaptive regret minimizer for adversarial MDPs. State-of-the-art MDPs online learning techniques rely on OMD-like (with KL regularizer) optimization updates, which make the no-interval regret property unattainable. Resorting to OGD-like updates and changing the theoretical analysis allows us to build an algorithm which is no-regret for every interval of the learning dynamic, thus having more guarantees in terms of stability.
> The aforementioned achievement may push the online learning research area to develop more stable regret minimizers even in complex environments such as Markov decision processes, since the classical static regret definition might not always make sense, especially when the environment is non-stationary such that there is no single fixed action that performs well in every interval of the horizon.
>
> To summarize, even if some results seem to be in common with [1], the theoretical analysis are quite independent. Indeed, as previously specified, the main theoretical effort of our work lies in showing the bound on the Lagrangian multiplier given the no-interval regret property of both the primal and the dual regret minimizers. This kind of analysis cannot be found in [1].
>
> For additional details on the other technical challenges faced in our paper and its novelty over prior works, we invite the Reviewer to read our answer to the third question.

---

> ### Author Response · Authors · 2023-11-15
>
> >The term 'best-of-both worlds' is somewhat misleading. In the context of bandits and RL, 'best-of-both-worlds' typically implies achieving  $\sqrt{T}$ regret in the adversarial setting versus $\log(T)$ regret in the stochastic setting. This paper only demonstrates a $\sqrt{T}$ rate in a fully stochastic setting when condition 2 holds. Furthermore, the 'best-of-both-worlds' literature in CMDPs is not discussed at all. Additionally, many algorithms can achieve zero constraint violations. Therefore it's unclear if this is the best achievable result.
>
> We agree with the Reviewer that the term 1best-of-both worlds' has a different meaning in the **unconstrained** online learning literature. Nevertheless, **in constrained online optimization the definition of 'best of both world' is the one given in our paper** (e.g, see ["The Best of Many Worlds: Dual Mirror Descent for Online Allocation Problems", 2021], Castiglioni et al. (2022a), and Castiglioni et al. (2022b)).
> We believe that our work is more connected with the area of constrained optimization in which our definition of best-of-both-worlds is standard.
> Nonetheless, we will clarify this aspect in the final version of the paper.
>
> As concerns the 'best-of-both-worlds' literature in CMDPs (according to the definition of 'best-of-both-worlds' used in constrained online optimization or according to definition commonly used in unconstrained MDPs), it is not discussed in the paper since, to the best of our knowledge, it does not exist. If the Reviewer can provide us some references, we are glad to add them in the final version of the paper. Indeed, our work is **the first to consider fully-adversarial constraints in CMDPs, and thus the first 'best-of-both-worlds' work**. In order to further clarify the distinction with the literature on 'best-of-both-worlds' algorithms in unconstrained online learning, we will add a discussion on papers belonging to the latter literature in the final version of the paper.
>
>
> Regarding achieving a $\mathcal{O}(\log(T))$ regret bound in the stochastic setting, it has been shown that $\mathcal{O}(\log(T))$ regret bounds are not achievable online optimization with constraints. Previous works on this topic (see, e.g., ["The Best of Many Worlds: Dual Mirror Descent for Online Allocation Problems", 2021]) show a lower bound $\Omega(\sqrt{T})$ on the regret even in simple instances. As a consequence, all the best-of-both-worlds algorithms for similar problems have a $\mathcal{O}(\sqrt{T})$ regret in the stochastic setting (see, e.g., ["The Best of Many Worlds: Dual Mirror Descent for Online Allocation Problems", 2021], ["A Unifying Framework for Online Optimization with Long-Term Constraints", 2022]).
> We acknowledge that there are some works that deal **only** with the stochastic setting (i.e., do not provide best-of-both-worlds guarantees) and provide $\mathcal{O}(\log(T))$ regret; this happens due to quite **strong** assumptions on the problem, such as assuming the the optimal solution is deterministic (see  ["Bandits with knapsacks beyond the worst case", 2021]).
>
> As concerns results achieving zero constraint violation, we believe that the Reviewer is referring to ["Learning Policies with Zero or Bounded Constraint Violation for Constrained MDPs'', 2021]. In this paper, the authors assume to know a strictly feasible policy to employ in the first episodes, namely, when the uncertainty about the environment is high.  We can further elaborate about this point if the Reviewer prefers to.
> In general, there are works that provide stronger results for stochastic or adversarial CMDPs, but these works have much stronger assumptions.

---

> ### Author Response · Authors · 2023-11-15
>
> > All the theoretical results are standard, and the use of projection to bound the dual variable for boundedness is not surprising.
>
> Most of the works on constrained MDPs (and constrained online optimization) exhibit quite similar approaches, and the difference in the algorithm and analysis are always a bit subtle.
> For example, comparing ["Exploration-Exploitation in Constrained MDPs", 2020] and ["Upper Confidence Primal-Dual Reinforcement Learning for CMDP with Adversarial Loss", 2020], it easy to see that both the papers employ primal-dual techniques. Similar reasoning holds for the constrained online learning literature, where ["Online Learning with Knapsacks: the Best of Both Worlds", 2022],  ["Adversarial bandits with knapsacks", 2022] and ["A unifying framework for online optimization with long-term constraints", 2023] employ almost the same algorithm. Nevertheless, these works appear in top venues. The reason why all these works might look similar at a first glance is that they usually present a generic primal dual template that is then instantiated with different subroutines. However, the specific subroutines and the analysis are where the real contribution of these works lies.
>
> We strongly believe that compared to the homogeneity of the algorithms and proof techniques of previous works, we can claim that our algorithms and analysis is quite different from previous works. For instance, **we are the first to use regret minimizers with the interval regret property in MDP problems, and, thus, to automatically prove a bound on the Lagrange multipliers when the constraints are adversarial**, while **the projection on the restricted Lagrangian space is used only in extreme scenarios, namely $\rho$ is particularly small**. We believe that this cannot be consider a trivial extension. In the following, we describe more in detail the contributions of our paper.
>
> We developed the first best-of-both-worlds algorithm for CMDPs which works without knowledge of the feasibility parameter $\rho$. We introduce a complexity condition (namely, Condition 2), which indicates how hard the underlying constrained optimization problem is. Related works, such as ("Exploration-Exploitation in Constrained MDPs", 2020), implicitly assumes the aforementioned condition, inserting the feasibility parameter $\rho$ directly in the regret bound.
>
> When the condition holds, we automatically prove a bound on the Lagrangian multiplier learning dynamic thanks to the no-interval regret properties guaranteed by both the primal and the dual regret minimizers. Related works in simpler single-state constrained optimization do not bound the multiplier employing the no-interval regret property, since they assume knowledge of the feasibility parameter $\rho$, which is not reasonable in practice. Given the bound, we show $\tilde{\mathcal{O}}(\sqrt{T})$ violation in both the stochastic and the adversarial setting, $\tilde{\mathcal{O}}(\sqrt{T})$ regret bound in the stochastic setting and fraction of the optimal rewards guarantee in the adversarial one.
>
> Differently from the state-of-the-art, we extended the analysis to cases where Condition 2 is not satisfied. Precisely, our algorithm guarantees sublinear regret and violation in the stochastic setting even in the hardest scenario. This result follows from the choice of instantiating the dual regret minimizer in a restricted decision space, namely $\mathcal{D}=[0,T^{1/4}]^m$, which guarantees the optimal trade-off between regret bound and violation (both of order $\tilde{\mathcal{O}}(T^{3/4})$).
>
> From an algorithmic perspective, our algorithm is the first in the CMDP research area to explicit instantiate two black-box regret minimizers in order to solve the underlying Lagrangian game. This makes our work more general, since both no-regret algorithms could be substituted with algorithms with the same properties. Currently, our primal algorithm is the only one for Online MDPs to achieve the weak no-interval regret property, but in future it could be updated with a more efficient version.
>
> Moreover, our algorithm is the first one to employ OGD on a decision space which is the occupancy measure space, while in the literature Multiplicative Weights is standard. This key modification helped us to achieve the weak no-interval regret property. The primal regret minimizer employs an adaptive learning rate, which depends on the maximum loss encountered in the learning dynamic, that is **not** standard in the Online MDP literature. This trick helped us to achieve better regret guarantees.
>
> Finally, our algorithm does not employ policy mixing techniques, nor exploration bonus ("Exploration-Exploitation in Constrained MDPs", 2020), which leads to a more understandable learning dynamics.

---

> ### Author Response · Authors · 2023-11-15
>
> > In the unconstrained case, several works have discussed bandit feedback and constraint violation without cancellation. However, the authors have not provided much insight into the challenges of removing these assumptions and studying the stronger setting.
>
> As it is usual in the online learning literature, both for single state and multi state environments, when a novel problem is proposed, it is first studied in the full-feedback setting; then, it is extended to the bandit setting, when only the realizations are observed (i.e. from Hedge to Exp3 ["A modern introduction to online learning", 2019] and, in case of MDPs, from ["Online convex optimization in adversarial Markov decision processes", 2019] to ["Online shortest path with bandit feedback and unknown transition function", 2019] ).
>
> Nevertheless, to extend our result to bandit settings, it is sufficient to develop a regret minimizer for MDPs with bandit feedback which guarantees no-interval regret.
> Then, the result will follow immediately from our analysis. Nevertheless, to develop no-interval regret minimizer for adversarial MDPs with bandit feedback is a non-trivial challenge. We believe that future work can build on this paper and design algorithms with less informative feedback.
>
> As concerns the constraints without the cancellation, primal-dual methods fails in achieving sublinear violation without cancellations (we can further discuss about the technical challenge). The algorithms which achieves sublinear violation without cancellation generally resort to confidence bound, which cannot work in the adversarial setting.
>
> >The regret bounds depend on condition 2. How is this related to the Slater condition, and which one is stronger? If they are not closely related, what happens if condition 2 doesn't hold but the Slater condition does? In such a case, your algorithm cannot achieve results of the same order as an algorithm using the Slater condition. How can we even say if it is the best of both worlds?
>
>
> Formally, Condition 2 is slightly stronger than the Slater's condition. However, all the previous works assume that $\rho$ is a **constant** and simply hide the dependence on $\rho$ in the regret bound. For instance, if the slackness parameter $\rho$ is really small, e.g., $1/T$, the regret bound is superlinear in $T$! (see, e.g., the regret bound without big O notation in ["Upper Confidence Primal-Dual Reinforcement Learning for CMDP with Adversarial Loss", 2021]).
> We extend this result and provide an analysis that holds even in degenerate cases in which $\rho$ is arbitrary small. If the Reviewer desires a classical (and weaker) regret bound, it is sufficient to consider only the proof of the case in which Condition 2 holds (even if the condition does not hold).
> In this way, we have a bound that depends on $\rho$ and $\sqrt{T}$, and assuming that $\rho$ is constant (since it can be hidden in the big-O notation as in previous works), a $\mathcal{O}(\sqrt{T})$ regret bound.
> To conclude, our results are actually stronger than the one in previous works when slackness parameter $\rho$ is very small and comparable otherwise.

---

> > ### Comment · Reviewer_KHRX · 2023-11-22
> >
> > I thank the authors and appreciate their detailed responses. You indicated that the main contribution is ensuring that the Lagrange multipliers remain bounded. It indeed can be achieved through the projection of the multipliers to me.
> >
> >  Regarding my concern about achieving the "best of both worlds," as mentioned in your paper, your algorithm can achieve optimal regret and constraint violation bounds. However, in my view, the results are far from optimal, given that good results have already been established in online convex optimization literature.
> >
> > In the stochastic case, all reward and cost functions are revealed after the first episode, which should simplify the problem compared to algorithms that only consider the stochastic setting. However, your results, even in the small $\rho$ case, show the regret and violation bound as $\zeta\sqrt{T}$. This may imply a larger dependence on T due to your assumption on $\rho$ (since $\zeta$ is defined in terms of $\rho$). Consequently, the results for the adversarial case are even worse. While I agree that you have an algorithm for both settings, I doubt we can claim it represents the best of both worlds, as this could be misleading. Readers might be initially excited by the abstract but could feel disappointed after reading the results.

---

> > > ### Author Response · Authors · 2023-11-22
> > >
> > > > You indicated that the main contribution is ensuring that the Lagrange multipliers remain bounded. It indeed can be achieved through the projection of the multipliers to me.
> > >
> > > You suggested to bound the Lagrangian Multiplier. On which set are you suggesting to project the Lagrangian Multiplier? Without any knowledge of the problem instance, the "best" thing is to project it on $T^{1/4}$, resulting in a suboptimal regret bound of $T^{3/4}$.
> > >
> > > > Regarding my concern about achieving the "best of both worlds," as mentioned in your paper, your algorithm can achieve optimal regret and constraint violation bounds. However, in my view, the results are far from optimal, given that good results have already been established in online convex optimization literature.
> > >
> > > All the bounds on our paper are provably optimal. As we explain in the previous message, our regret is $\sqrt{T}$ under Slater condition, matching the regret bounds of previous works. We encourage the reviewer to look at previous work to figure out that when the Slater parameter is small, the regret bound of **ANY** primal-dual method blows out. Moreover, our bound are provably optimal also in the adversarial setting (as we wrote in the previous message).
> > >
> > > > In the stochastic case, all reward and cost functions are revealed after the first episode
> > >
> > > There is a misunderstanding about our feedback model. We observe a sample from the reward and cost functions. Thus, the true reward function cannot be learned at the first episode (and neither at the last). Indeed, also under full-feedback even a simple expert problem (MAB with full-feedback) has a lower bound $\sqrt{T}$ on the regret.

---

### Official Review · Reviewer_r196 · 2023-11-01

**Soundness:** 3 good
**Presentation:** 3 good
**Contribution:** 3 good
**Rating:** 6
**Confidence:** 3

**Summary:**

This paper considers the online learning problem of episodic constrained MDP for T rounds, and provides the first best-of-both-worlds algorithm for CMDPs with long-term constraints. In other words, the proposed algorithm matches state-of-the-art regret and constraint violation bounds for settings in which constraints are selected both stochastically and adversarially. Specifically, the long-term constraints setting allows the agent to violate the constraints in a given episode while the cumulative violation is controlled by growing sublinearly in the number of episodes.

**Strengths:**

1. The online learning problem of CMDP is a fresh and important setting. And this paper achieves the first best of both worlds guarantee of such setting.
2. The analysis of the parameter $\rho$ seems to be able to be applied in other best of both worlds problem.
3. The mathematical proof (though I just skimmed several lemmas) is rigorous.

**Weaknesses:**

This paper does not have any specific weaknesses.

**Questions:**

1. Is it possible to achieve the best of both worlds guarantee for bandit feedback? Any conjecture?
2. Is it possible to achieve the best of both worlds guarantee with logarithmic stochastic regret bound?

---

> ### Author Response · Authors · 2023-11-15
>
> > Is it possible to achieve the best of both worlds guarantee for bandit feedback? Any conjecture?
>
> We conjecture that it is possible to achieve the best of both worlds guarantee for bandit feedback. Precisely, to extend our result to bandit settings, it is sufficient to find a regret minimizer for adversarial MDPs with bandit feedback which guarantees no-interval regret. Then, the result will follow immediately from our analysis. Nevertheless, developing a no-interval regret minimizer for adversarial MDPs with bandit feedback is a non-trivial challenge. We believe that future work can build on this paper and design algorithms with less informative feedback.
>
> >Is it possible to achieve the best of both worlds guarantee with logarithmic stochastic regret bound?
>
> We thank the Reviewer for the opportunity to clarify this fundamental aspect. In the setting we propose, it is not possible to achieve $\mathcal{O}(\log(T))$ regret and violation in the stochastic setting. Indeed, previous works (see, e.g., ["The Best of Many Worlds: Dual Mirror Descent for Online Allocation Problems", 2021]) show a lower bound $\Omega(\sqrt{T})$ on the regret even in simple instances. As a consequence, all the best of both worlds algorithms for similar problems have a $\mathcal{O}(\sqrt{T})$ regret in the stochastic setting (see, e.g., ["The Best of Many Worlds: Dual Mirror Descent for Online Allocation Problems", 2021], ["A Unifying Framework for Online Optimization with Long-Term Constraints", 2022]).
>  We will better highlight the difference with this other best of both worlds definition in the final version of the paper. We acknowledge that there are some works that deal **only** with the stochastic setting (i.e., do not provide best of both worlds guarantees) and provide $\mathcal{O}(\log(T))$ regret; this happens due to quite **strong** assumptions on the problem, such as assuming that the the optimal solution is deterministic (see  ["Bandits with knapsacks beyond the worst case.", 2021]).

---

### Official Review · Reviewer_bJQa · 2023-11-05

**Soundness:** 3 good
**Presentation:** 3 good
**Contribution:** 3 good
**Rating:** 6
**Confidence:** 3

**Summary:**

This paper studies the Markov devision processes with long-term constraints.  The paper formulates the constrained MDP as a linear programming problem based on the occupancy measure.  The reward and constraint matrix can be adversarially or stochastically chosen by the environment. A primal-dual algorithm is proposed to learn the policy under the long-term constraints under both adversarial and stochastic settings. The paper proves that for both adversarial and stochastic settings, the regret and constraint violation are all sublunar with $T$.

**Strengths:**

+ This paper considers CMDP with adversarial reward and constraints, which was not considered in other literature.

+ A primal-dual algorithm is proposed to achieve sub-linear regret and constraint violation for CMDP under both stochastic and adversarial cases. The design of confidence set presents new challenges.

+ The paper is well-written and easy to follow.

**Weaknesses:**

- The primal-dual framework is widely used to solve CDMP or constrained online optimizations. The sublinear regret and constraint violation can be proved in many adversarial settings of constrained online optimization [1].  Can the authors discuss more on the challenges to achieve provable regret and constraint violation bound in the considered setting?

- The paper only considers the tabular MDP which is simple. Can the authors discuss the possible generalization to continuous actions and/or continuous states?

- Although this paper has a theory taste, it would be better to have empirical results to evaluate the proposed algorithms.

[1] Neely, Michael J., and Hao Yu. "Online convex optimization with time-varying constraints." arXiv preprint arXiv:1702.04783 (2017).

**Questions:**

- The adversarial setting is about the adversarial reward and constraint matrices, but would it be possible to design an algorithm for adversarial transition kernel?

---

> ### Author Response · Authors · 2023-11-15
>
> > The primal-dual framework is widely used to solve CDMP or constrained online optimizations. The sublinear regret and constraint violation can be proved in many adversarial settings of constrained online optimization [1]. Can the authors discuss more on the challenges to achieve provable regret and constraint violation bound in the considered setting?
>
> The fundamental difference of our paper with respect to [1] (and similar works) is that our baseline for the computation of the cumulative regret is **stronger**. Indeed, our algorithm compares itself with the best strategy that is safe in hindsight (see, definition of the regret $R_T$ in Section 2.4 in our paper). On the contrary, [1] uses as baseline a strategy that is safe at **every** round, thus being possibly **extremely** loose in terms of performances (precisely, in the 'Formulation' section, [1] states: "The above goal compares against fixed-decision vectors $x\in A$ that make all constraint functions nonpositive for all slots $t$").
> As proved by Mannor et al. (2009), it is not possible to achieve both sublinear regret and sublinear violation in the adversarial setting, **when one considers our baseline**. Thus, the best achievable result is to guarantee sublinear violation and a fraction of the optimal rewards. Furthermore, let us remark that if we adopt the weaker baseline of [1], it is possible to show that even our algorithm achieves sublinear regret and violation in the adversarial setting (we can provide additional technical details if the Reviewer needs them).
>
> >The paper only considers the tabular MDP which is simple. Can the authors discuss the possible generalization to continuous actions and/or continuous states?
>
> Our algorithm (as it is standard in the online learning in Adversarial MDPs literature, see, e.g., ["Online convex optimization in adversarial Markov decision processes", 2019], ["Online Stochastic Shortest Path with Bandit Feedback and Unknown Transition Function", 2019], ["Learning adversarial MDPs with bandit feedback and unknown transition", 2019]) resorts to a Linear Program (LP) formulation of (constrained) Markov decision processes. Indeed, the generalization to continuous actions and/or continuous states would lead to LPs with an infinite number of constraints, thus not solvable in polynomial time. Nevertheless, notice that (constrained) MDPs with continuous state/action set are generally studied from an experimental point of view, since their theoretical analysis is highly non-trivial without making strong assumptions as in linear Markov decision processes or similar settings.
>
> >Although this paper has a theory taste, it would be better to have empirical results to evaluate the proposed algorithms.
>
> We agree with the Reviewer that numerical experiments are always beneficial. Nevertheless, in the research area this work lies, namely online decision-making with long-term constraints, experiments are almost never included, see, e.g., ["Online convex optimization with stochastic constraints", 2017], ["A Unifying Framework for Online Optimization with Long-Term Constraints", 2022]. This is also true for the online learning in adversarial Markov decision process research area, see, e.g., ["Online convex optimization in adversarial Markov decision processes", 2019], ["Online Stochastic Shortest Path with Bandit Feedback and Unknown Transition Function", 2019], ["Learning adversarial MDPs with bandit feedback and unknown transition", 2019]. Please notice that, despite the lack of experiments, all the aforementioned works were published in top AI conferences, namely, venues similar to ICLR.
>
> >The adversarial setting is about the adversarial reward and constraint matrices, but would it be possible to design an algorithm for adversarial transition kernel?
>
> In general, there exists many negative results related to MDPs with adversarial rewards and adversarial transitions. For example, in such a setting, it is well-known that it is not possible to achieve sublinear regret (see, ["Online learning in unknown markov games", 2021]), without making strong assumptions on the total variation of the transitions, thus not being fully adversarial. We conjecture that, without similar (strong) assumptions, no positive results (in terms of both regret and violations) are achievable in the same setting, with the additional complexity of dealing with constraints.

---

### Meta-Review · Area_Chair_1cgm · 2023-12-05

**Metareview:**

This paper studies the Markov devision processes with long-term constraints. The reward and constraint matrix can be adversarially or stochastically chosen by the environment. Regret and Constraint violation bounds are studied.

Strength:
The key strength is that this is first work for CMDPs with adversarial rewards and constraints.

Weakness:
1. The approach uses ideas from (Castiglioni et al., 2022) which has similar results for optimization problem.
2.  Standard OMD cannot have this "no-interval regret" properties, and the authors resolve this by the gradient descent analysis. However, a standard way to achieve no-interval regret in OMD is to perform weight update on a truncated action space (lower bound away from 0). It would be good argue why such standard idea does not work in their setting. This may limit the mentioned technical novelty, if the standard approach works.
3. There are works on zero constraint violation in all of average reward, discounted reward, and episodic cases. Further, such results exist for episodic, both in model-based and model-free setups. The authors should try to see if they can also obtain the zero constraint violation result in their case.

**Justification For Why Not Higher Score:**

Concerns have been raised regarding the paper's limited technical novelty.

**Justification For Why Not Lower Score:**

N/A

---

### Decision · Program_Chairs · 2024-01-16

Reject